# Increasing Resolution and Accuracy in Sub-Seasonal Forecasting through 3D U-Net: the Western US

Jihun Ryu[1,2], Hisu Kim[3], Shih-Yu (Simon) Wang[4], and Jin-Ho Yoon[1*]

[1]School of Environment and Energy Engineering, Gwangju Institute of Science and Technology, Gwangju, South Korea
[2]Department of Plants, Soils and Climate, Utah State University, Logan, UT, USA
[3]School of Electrical Engineering and Computer Science, Gwangju Institute of Science and Technology, Gwangju, South Korea
[4]Department of Agronomy, Kasetsart University, Bangkok, Thailand

**Correspondence:** Jin-Ho Yoon (yjinho@gist.ac.kr)

**Abstract.** Sub-seasonal weather forecasting is a major challenge, particularly when high spatial resolution is needed to capture complex patterns and extreme events. Traditional Numerical Weather Prediction (NWP) models struggle with accurate forecasting at finer scales, especially for precipitation. In this study, we investigate the use of 3D U-Net architecture for post-processing sub-seasonal forecasts to enhance both predictability and spatial resolution, focusing on the western U.S. Using the ECMWF ensemble forecasting system (input) and high-resolution PRISM data (target), we tested different combinations of ensemble members and meteorological variables. Our results demonstrate that the 3D U-Net model significantly improves temperature predictability and consistently outperforms NWP models across multiple metrics. However, challenges remain in accurately forecasting extreme precipitation events, as the model tends to underestimate precipitation in coastal and mountainous regions. While ensemble members contribute to forecast accuracy, their impact is modest compared to the improvements achieved through downscaling. The model using the ensemble mean and only the target variables was most efficient. This model improved the pattern correlation coefficient for temperature and precipitation by 0.12 and 0.19, respectively, over a 32-day lead time. This study lays the groundwork for further development of neural network-based post-processing methods, showing their potential to enhance weather forecasts at sub-seasonal timescales.

## 1 Introduction

Sub-seasonal forecasting based on numerical weather prediction (NWP) models has made significant advances over the past few decades, with the ability to predict extreme events such as heat waves up to four weeks in advance (Ardilouze et al., 2017; Vitart and Robertson, 2018). However, limitations still exist, which have led to increasing interest in deep learning models as alternative approaches for weather forecasts. Some models directly generate the forecasts from the input data. Weyn et al. (2021) aimed to provide ensembles similar to those in NWP systems. Two deep learning models, GraphCast and Pangu, have outperformed NWP in weather and medium-range forecasts, from 1 day to 10 days (Bi et al., 2023; Lam et al., 2023). More recently, deep learning models such as Fuxi-S2S have been reported to surpass NWP in sub-seasonal forecasting (Chen et al.,

2024). Among them, GraphCast does not provide precipitation forecasting, while these models only generate deterministic forecasts and struggle with predicting extreme weather events (Olivetti and Messori, 2024).

On the other hand, post-processing NWP outputs have also been explored as a means of improving forecast accuracy (Wool-nough et al., 2024). In recent years, neural network-based post-processing methods have gained traction. The U-Net architecture has been widely utilized for weather forecast post-processing due to its ability to capture fine details through contracting and expanding layers (Horat and Lerch, 2024; Faijaroenmongkol et al., 2023; Deng et al., 2023; Xin et al., 2024). U-Net has also shown potential in probabilistic forecasting for sub-seasonal predictions (Horat and Lerch, 2024). Furthermore, U-Net was employed to correct biases in seasonal precipitation forecasts in Thailand (Faijaroenmongkol et al., 2023).

Moreover, generating high-resolution NWP outputs demands significant computational resources, so deep learning has been applied to downscale sub-seasonal forecasts and simultaneously improve predictability efficiently. For example, studies in wildfire weather forecasting in the western United States have successfully downscaled predictions to the county level (Son et al., 2022). Another example is the improved predictability and downscaling of temperature and precipitation in China, achieved by using a weighted combination of multiple models based on a U-Net (Xin et al., 2024).

**Table 1.** Comparison of the proposed method with previous post-processing studies.

| Study | Type | Input | | Output | |
|---|---|---|---|---|---|
| | | Ensemble | Variable | Lead time | Model by lead time |
| Our study | post-processing | individual member | used additional variables | 0 – 32 days, daily | One model for forecast period |
| Rasp and Lerch (2018) | post-processing | mean, std | used additional variables | 48h | Lead time specific model |
| Schulz and Lerch (2022) | post-processing | mean, std, individual member | used additional variables | 0–21 h, hourly | Lead time specific model |
| Höhlein et al. (2024) | post-processing | individual member | used additional variables | wind gust: 6h, 12h, 18h temperature: 24h, 72h, 120h | Lead time specific model |
| Horat and Lerch (2024) | post-processing | mean | used additional variables | temperature: 3–4W, 5–6 W mean precipitation: 3–4W, 5–6 W accumulate | Lead time specific model |

A key consideration in these studies is the selection of input data. Some studies use only target variables, meaning the same variable is used as both input and target, such as using ECMWF precipitation as input and PRISM precipitation as the target (Xin et al., 2024), while others use a broader set of additional variables (Horat and Lerch, 2024; Weyn et al., 2021). The extent to which inputs significantly affect sub-seasonal forecasting remains undetermined and case-sensitive. Even though studies on weather forecasts have found that additional variables play a limited role in temperature forecasting, they have demonstrated improvements in wind gust predictions (Rasp and Lerch, 2018; Schulz and Lerch, 2022). Additionally, attempts to utilize each ensemble member of the NWP for U-Net training resulted in only marginal improvements in weather forecasting accuracy. (Höhlein et al., 2024). Prior evaluations of predictor sets and ensemble usage have largely been limited to short lead times ($\leq 5$ days) and single valid times (Rasp and Lerch, 2018; Schulz and Lerch, 2022; Höhlein et al., 2024), probing predictability at an instant (Table 1) (Rasp and Lerch, 2018; Schulz and Lerch, 2022; Höhlein et al., 2024). In contrast, we target sub-seasonal forecasting by supplying sequences of forecast lead times to encode, thereby extending previous findings to lead times longer.

This study enhances predictability in the Western United States through the 3D U-Net-based post-processing that encodes temporal information via forecast lead times and downscaling forecasts to higher spatial resolutions. In doing so, we identify

the role played by ensemble members and additional variables in enhancing predictability and investigate whether downscaling with neural networks leads to meaningful improvements at smaller scales such as the county level. Section 2 describes the data, our 3D U-Net architecture which uses three-dimensional convolution to capture spatial and temporal features, pre-processing, and evaluation metrics, while Section 3 discusses the results and analysis. Lastly, conclusions are presented in Section 4.

## 2 Data and Methodology

### 2.1 Data

This study employs two primary datasets: the European Centre for Medium-Range Weather Forecasts (ECMWF) real-time perturbed forecasts and the Parameter-elevation Regressions on Independent Slopes Model (PRISM) dataset. First, as the ECMWF forecast model from the sub-seasonal to seasonal (S2S) prediction project continues to evolve, providing an increasing number of ensemble members, forecast periods, and forecast cycles, we select the $1.5° \times 1.5°$ resolution (approximately 120 km × 120 km over the study region), 50 ensemble perturbation forecasts, twice-weekly forecast cycles, and 32-day lead times to match the earliest version of the ECMWF model (Roberts et al., 2018). The 2 m temperature and total column water are provided as daily averaged, while the other variables are available with 6-hourly frequency. We utilize forecasts from CY40R1 to CY48R1, covering the period from January 2015 to December 2023. For detailed information on each version of the model, please refer to the ECMWF model archive: https://confluence.ecmwf.int/display/S2S/ECMWF+Model. These forecasts span weather to sub-seasonal time scales, offering a comprehensive range of meteorological variables essential for our neural network post-processing model. Next, we utilize the daily PRISM dataset, developed by Oregon State University, which provides high-resolution climate data for the United States (Daly et al., 2008) for the sake of model validation and high-resolution reference data. PRISM offers grid estimates of variables including temperature, precipitation, and elevation at a fine spatial resolution of $0.042° \times 0.042°$ (approximately 4 km). Only data from January 2015 to January 2024 are used, corresponding to the period of ECMWF forecasts utilized in this study. An overview of the dataset is provided in Table S1.

We chose the Western United States because it is a diverse region, ranging from coastal areas to high mountain ranges, and the importance of water management emerges in the face of hydrological changes driven by the climate crisis (Siirila-Woodburn et al., 2021). To evaluate the model's performance at finer spatial scales, we select five diverse regions in the Western United States, each representing different climatological socio-economic characteristics. These regions include three highly populated urban areas and two important agricultural zones. In detail, we choose (1) San Francisco, California, a major high-populated metropolitan area with a unique coastal climate; (2) Orange County, California, known for its citrus farming and Mediterranean climate; (3) the area around the Great Salt Lake in Utah, which combines high population density with a distinctive lake-effect climate; (4) Seattle, Washington, representing the Pacific Northwest's urban environment and maritime climate; and (5) a vast wheat farming region in eastern Washington, exemplifying the inland agricultural areas of the West (Fig. S1).

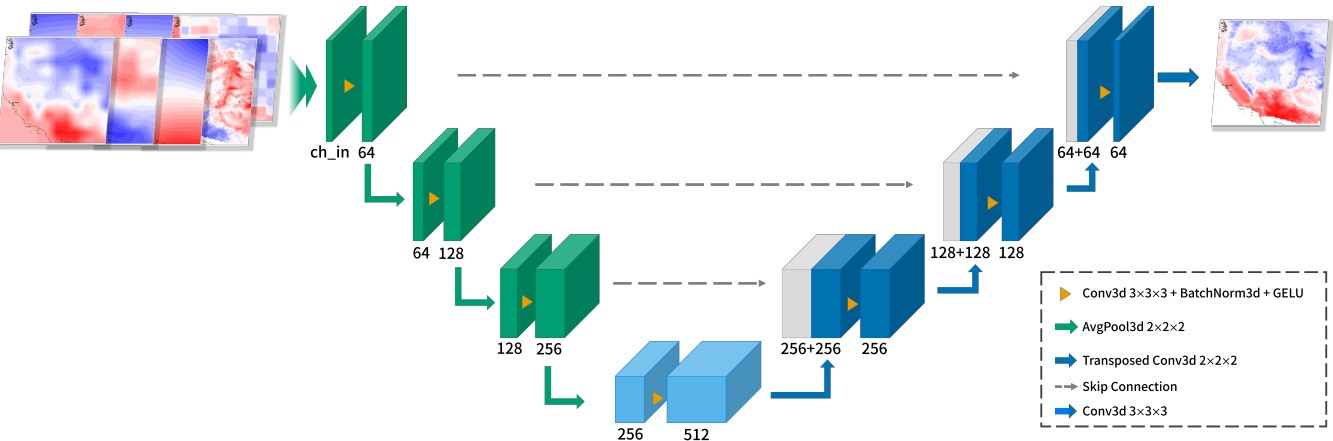

**Figure 1.** Schematic of the 3D U-Net architecture adapted for weather forecast post-processing. The model consists of a contracting path (left, green), an expanding path (right, blue), and a bottleneck layer (center, teal), with skip connections (dashed gray arrows) preserving spatial information. Operations between layers are described in the dashed box on the right corner.

## 2.2 3D U-Net Architecture

The post-processing approach utilizes the U-Net architecture, originally proposed by Ronneberger et al. (2015) for biomedical
80    image segmentation. The U-Net is particularly well-suited for our task of enhancing sub-seasonal forecasts due to its ability to capture multi-scale features and preserve spatial information through skip connections (Horat and Lerch, 2024). We add the height dimension to implement the 3D U-Net structure, accounting for the temporal continuity inherent in meteorological variables such as temperature and geopotential height, as shown in Fig. 1. In this framework, lead time is treated as the vertical dimension. This allows the model to utilize information from shorter lead times, which typically exhibit higher predictive skill,
85    to improve sub-seasonal forecasts. Additionally, this structure enables the generation of daily forecasts, in contrast to traditional approaches that rely on weekly averages, thereby providing a finer temporal resolution for downstream applications.

     The 3D U-Net structure consists of a contracting path (encoder) and an expanding path (decoder), connected by a bottleneck layer. Our implementation features three contracting-expanding cycles, optimized for the spatial scales relevant to sub-seasonal forecasting. The contracting path progressively reduces spatial dimensions (moving from fine to coarse) while increasing
90    feature channels, allowing the model to capture broader contextual information. Conversely, the expanding path restores spatial resolution (from coarse to fine), enabling precise localization of weather patterns. In short, this structure concatenates feature maps from the contracting path to the expanding path so that the model retains fine-grained spatial information that might otherwise be lost during downsampling.

     We train the model using ECMWF forecast fields as input and high-resolution PRISM reanalysis dataset as the target output.
95    To investigate the impact of ensemble forecasting on post-processing performance, we conduct experiments with different combinations of ensemble members: Using only the first ensemble member (E01), utilizing all 50 ensemble members (E50),

**Table 2.** Variable list for temperature and precipitation. Additional variable's order reflects the correlation coefficient, high to low.

| Target | Additional Variable |
|---|---|
| Temperature | t2m, u500, z200, v200, z500, mslp, topo, tcw |
| Precipitation | pr , mslp, z500, z200, u850, tcw, v500, v10 |

and employing the mean of all 50 ensemble members (E50M). Further, we explore the impact of input variable selection on model performance by testing configurations with varying numbers of meteorological additional variables (V1, V2, V4, V8). This exploration aims to determine whether incorporating additional variables beyond the target variable could enhance the model's predictive capabilities.

In our specific implementation, we integrate the ensemble members and variables into a channel, utilizing a 3D U-Net structure with forecast lead time, latitude, and longitude as the dimensions. The forecast period ranges from 1 day to 32 days ahead, with a longitude range of $235.5°$ to $253.25°$ and a latitude range of $31.25°$ to $49°$, consisting of 72 grid points in each. Based on the forecast start date, the training period spans from January 2015 to December 2020, the validation period from March 2021 to February 2022, and the test period from January 2023 to December 2023. For example, the E50 V8 configuration has 400 input channels, while the E50M V2 configuration has 2 input channels. In the model, the input dimensions are referred to as height, width, and depth, corresponding to lead time, latitude, and longitude, with sizes of 32, 72, and 72, respectively.

The 3D U-Net model was trained for 100 epochs using the adam-optimizer with an initial learning rate of 1e-4 and a batch size of 11, selected based on GPU memory limitations (Kingma and Ba, 2017). The network architecture consists of three encoding and decoding blocks, each composed of 3D convolutional layers with 3×3×3 kernels. Average pooling was used for downsampling in the encoder, and transposed convolution was used for upsampling in the decoder. The GeLU activation function was applied after each convolutional layer. To prevent overfitting, we applied early stopping based on validation loss with a patience of 10 epochs. The loss function combines mean squared error (MSE) and spatial pattern correlation, with equal weighting assigned to both components. We chose this combination because each metric emphasizes a different aspect of prediction performance. MSE evaluates the model's ability to reproduce the absolute magnitude of values, while spatial pattern correlation captures the fidelity of the overall spatial distribution, which is particularly important in sub-seasonal forecasting. All configurations were selected through trial-and-error experiments to ensure training stability and generalization capability. These details have been incorporated into the main manuscript for transparency.

## 2.3 Pre-processing

To assess the sensitivity of the additional variables used in the learning process, we select 16 variables: 2 m temperature, precipitation, total column water (tcw), mean sea level pressure (mslp), 10 m u-wind (u10), 10 m v-wind (v10), elevation, and geopotential height (z), along with u-wind (u) and v-wind (v) at the 850 hPa, 500 hPa, and 200 hPa levels. This includes variables representing large-scale circulation at four vertical levels: near surface, lower, mid, and upper troposphere. The tcw

was included to capture atmospheric rivers affecting precipitation in the western US. Elevation was included for its known benefit in temperature bias correction (Rasp and Lerch, 2018).

The dataset is split into two pre-processing groups, one being precipitation and tcw, and the other being topography and the remaining atmospheric variables. For precipitation and tcw, any negative values are set to zero, as they are non-physical for these types of data. We then apply conservative interpolation, a method that preserves physical quantities like mass or energy during spatial grid adjustments, to ensure the accurate preservation of values during spatial adjustments. For the remaining variables, linear interpolation was applied. All datasets were interpolated to the $0.25° \times 0.25°$ latitude-longitude grid for model input, with PRISM data downscaled from $0.042° \times 0.042°$ and ECMWF forecasts upscaled from $1.5° \times 1.5°$. Based on the fact that predictability can be evaluated using the mean state (Ryu et al., 2024), we calculate the mean state of each additional variable across both weather and sub-seasonal timescales. The spatial pattern correlation coefficient between the mean state of each additional variable and that of the target variable is then computed. The absolute values of these correlations are averaged across the two timescales, and the variables are ranked accordingly. Rankings are shown above each bar in Fig. S2. The top eight variables for each target are selected for use in the 3D U-Net model, as summarized in Table 2.

The interpolated dataset is further processed for input into deep learning models. For precipitation and tcw, following Aich et al. (2024), we applied a transformation to compress the wide range of precipitation values and facilitate stable, efficient model training. To handle zero values, we added 1 to the data and applied a log10 transformation. The transformed data is then standardized by calculating the mean and standard deviation, making it suitable for use in the 3D U-Net architecture. For the other variables, we follow standardization by computing the mean and standard deviation, similar to the pre-processing approach used in GraphCast (Lam et al., 2023). This normalization step ensures that all variables are prepared for efficient training in the 3D U-Net model.

## 2.4 Evaluation Metrics

Intending to assess the performance of our 3D U-Net-based post-processing model comprehensively, we employ the following three key evaluation metrics: pattern correlation (Eq. S1), root mean square error (RMSE) (Eq. S2), and $E_{pre}$ (Eq. 1) (Ryu et al., 2024). Pattern correlation evaluates the model's ability to reproduce the spatial distribution of temperature and precipitation fields, while RMSE quantifies the average magnitude of forecast errors at each grid point. Both metrics are commonly selected to evaluate sub-seasonal predictions. Lastly, we incorporate the $E_{pre}$ metric, which builds upon the concept of Taylor diagrams and has been utilized in several studies for evaluating forecast performance (Ryu et al., 2024; Wang et al., 2021; Yang et al., 2013). This metric offers a comprehensive assessment by integrating both the variance ratio and the correlation between predictions and observations. We measure $E_{pre}$ per lead time by averaging values of all initial dates.

$$E_{pre} = \frac{1}{N} \sum_{i=1}^{N} \log \left[ \frac{\left( \frac{\sigma_{obs,i}}{\sigma_{pre,i}} + \frac{\sigma_{pre,i}}{\sigma_{obs,i}} \right)^2 (1 + r_0)^4}{4 \left(1 + r_i\right)^4} \right] \tag{1}$$

Here, $\sigma_{obs,i}$ and $\sigma_{pre,i}$ denote the standard deviations of observed and predicted values respectively. $r_0$ represents an ideal correlation (set to 1), and $r_i$ is the actual correlation at time step $i$. $N$ stands for the number of initial dates. The $E_{pre}$ metric is

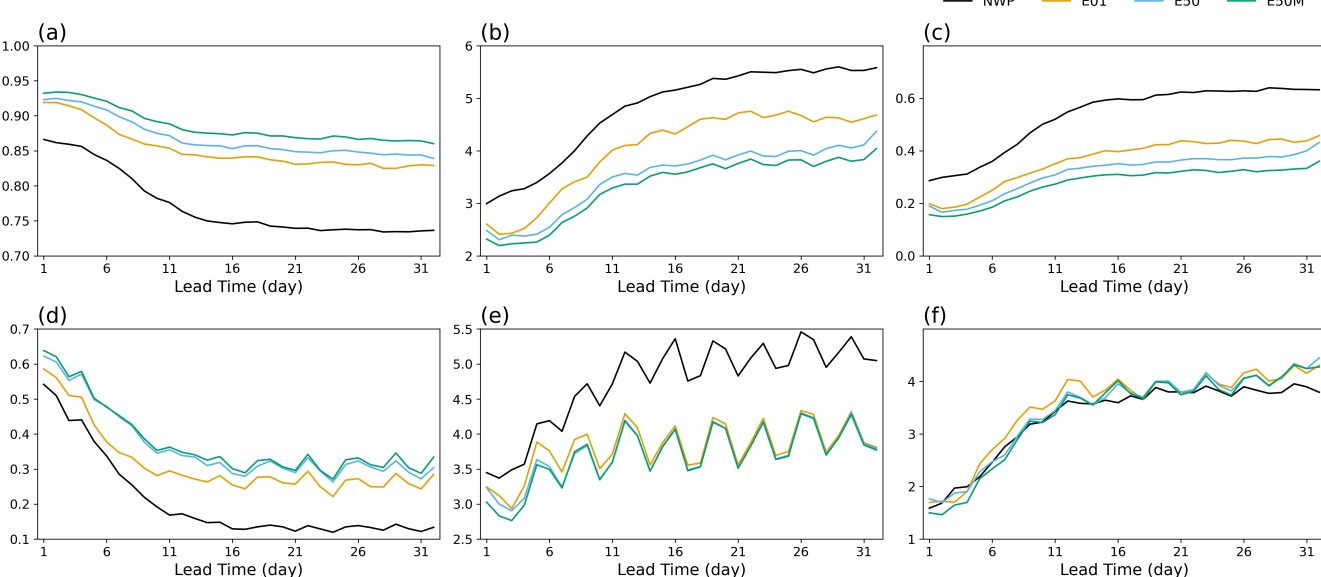

**Figure 2.** Ensemble sensitivity benchmark scores for the Western U.S., comparing NWP and 3D U-Net models (E01, E50, E50M) for temperature (top row) and precipitation (bottom row) forecasts over 32 days. Columns show (a, d) pattern correlation, (b, e) RMSE, and (c, f) $E_{pre}$, respectively.

designed to yield a value of 0 for perfect predictions, with increasing values indicating greater discrepancies between forecasts and reanalysis dataset. By incorporating both spread and accuracy considerations, this metric proves particularly valuable for evaluating the nuanced performance of ensemble predictions in sub-seasonal forecasting contexts.

## 3 Results and Discussion

### 3.1 Role of Ensemble and Variables

The performance of the 3D U-Net model, compared to traditional NWP forecasts, was evaluated across twelve cases combining three ensemble configurations and four input variable sets (Fig. S3). The 3D U-Net consistently outperformed the raw NWP forecasts across three evaluation metrics, except for $E_{pre}$ in precipitation. Statistical tests comparing each model's evaluation metrics with those of the NWP baseline showed that, apart from the $E_{pre}$ metric for precipitation, the improvements were significant. For precipitation $E_{pre}$, the results were mixed: five models (E01 V4, E50 V2, E50 V8, E50M V1, and E50M V2) showed no significant improvement, while seven models exhibited significant degradation.

Before conducting a detailed analysis of the results, we examined the potential for seasonal bias and the performance by land cover type. Our findings show improvements in all seasonal evaluation metrics for both temperature and precipitation, except for precipitation $E_{pre}$ in spring and summer (Figs. S4 and S5). This suggests that the enhanced performance is not

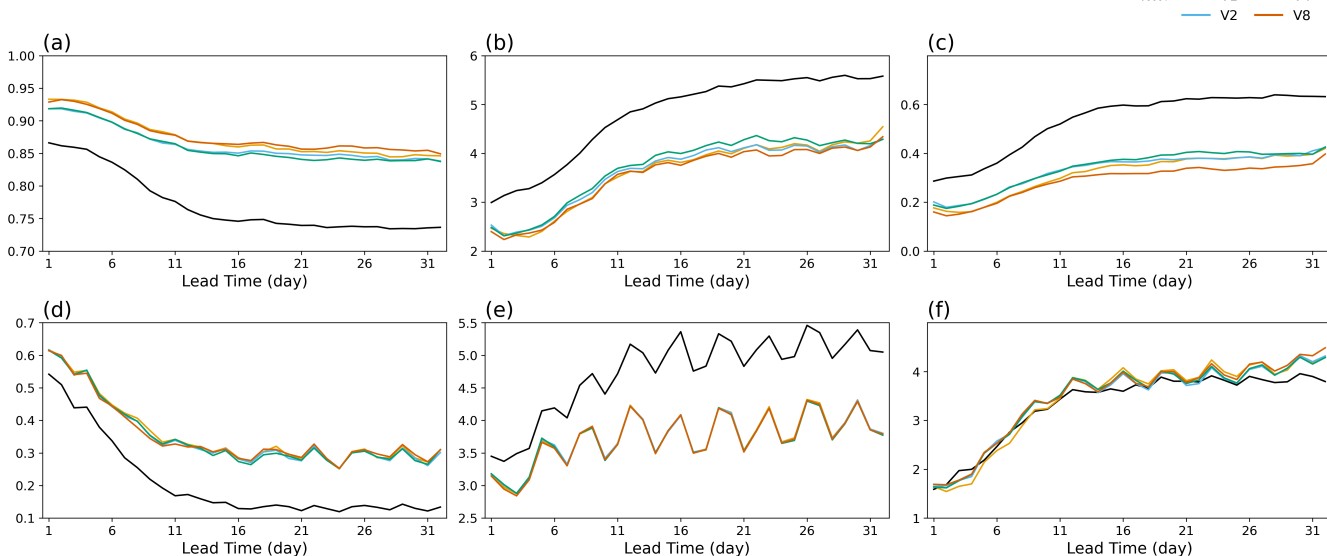

**Figure 3.** Additional variable sensitivity benchmark scores for the Western U.S., comparing NWP and 3D U-Net models with varying input variables (V1, V2, V4, V8) for temperature and precipitation forecasts over 32 days. Layout is the same as in Fig. 2.

simply due to the model converging toward the seasonal mean across all seasons. Rather, the improvements reflect the model's ability to capture relevant patterns within each season. Additionally, we analyzed model performance by land cover type using the National Land Cover Database (NLCD). The western United States is dominated by three land cover classes, Shrub/Scrub, Evergreen Forest, and Grassland/Herbaceous, which collectively cover over 80 percent of the study area (Fig. S6). Our analysis focused on these classes and found consistent performance patterns across all three (Figs. S7 and S8).

We then grouped experiments with the same input variables and ensemble configurations to assess the role of auxiliary variables and ensemble structure, for example, averaging E50 V1, E50 V2, E50 V4, and E50 V8 for the E50 group. Fig. 2 illustrates benchmark scores with respect to the ensemble configurations. In temperature predictions, E50M (see Section 2.2) shows the best performance and E01 is the most deficient in all metrics. Precipitation predictions also exhibit analogous operational characteristics: nonetheless, E50 and E50M exhibit significantly aligned trajectories and the overall disparity among all configurations has diminished in both pattern correlation and RMSE metrics. In contrast, $E_{pre}$ for precipitation does not show significant differences between NWP, likely due to limitations in precipitation variance. Results in Fig. 2 interestingly imply that E01 proved insufficient for effective learning by only the first ensemble member, resulting in performance lagging behind the other ensemble configurations. However, the performance difference between using E50 and E50M was negligible. To support these findings, we conducted a complementary experiment trained and tested with ERA5 data and tested on 2022 forecasts. The results indicate that ensembles of 10 and 20 members achieved performance comparable to E50 (Fig. S9). This suggests that while post-processing significantly improves forecast skill, the benefits of increasing ensemble members beyond the mean are limited for both temperature and precipitation predictions in the current setting. This is consistent with previous

research that ensemble spread plays a limited role in improving weather forecast accuracy, and these findings suggest that this limitation extends to sub-seasonal forecasts as well (Höhlein et al., 2024). In other words, using the ensemble mean could be sufficient for achieving optimal performance with the 3D U-Net model.

Our current approach produces deterministic forecasts and therefore cannot fully represent the uncertainty that NWP ensembles are designed to capture. To address this limitation, future work could consider several directions. One option is to train the 3D U-Net to generate probabilistic forecasts, for example via quantile regression with a pinball loss or by predicting parametric distributions (*e.g.*, Gaussian) optimized with the Continuous Ranked Probability Score (CRPS)(Hersbach, 2000; Gneiting and Raftery, 2007). Another is to evaluate the reliability and the relationship between spread and skill using Brier scores, rank histograms, and calibration methods such as isotonic regression. Finally, more advanced avenues could include modifying the network to produce its own ensemble or adopting Bayesian deep learning frameworks.

The impact of input variables on model performance is further explored in Fig. 3, which represents the averages of E01, E50, and E50M. The 3D U-Net models consistently outperform NWP across all lead times for both temperature and precipitation forecasts. This superiority reinforces the robustness of the neural network approach. Specifically, V8 shows significant improvements over V2 and V4 in all temperature metrics, but performs similarly to V1. This may be attributed to the inclusion of altitude, which has been shown to be one of the most important variables in temperature post-processing (Rasp and Lerch, 2018). However, for precipitation forecasts, V1, V2, V4, and V8 reveal insignificant variations in terms of their predictability scores. The $E_{pre}$ values for precipitation exhibited comparable patterns to those observed in Fig. 2(f), attributable to analogous underlying mechanisms. An intriguing observation is that the performance differences among the 3D U-Net models with varying numbers of input variables are minimal for both target variables. This contrasts with prior research, which has suggested that additional variables contribute to forecast improvement (Schulz and Lerch, 2022). However, our finding is consistent with studies that indicate additional variables may contribute only marginally or in a limited role, particularly when used mean state (Rasp and Lerch, 2018; Höhlein et al., 2024). This indicates that increasing the number of additional variables in the 3D U-Net model does not significantly enhance its ability to extract relevant information or improve forecast skills in this context. Such a result challenges the conventional wisdom that more input data invariably leads to better predictions, and suggests that the 3D U-Net architecture in the current setting may be efficiently capturing the most relevant features for the prediction even with a limited set of input variables. Thus we use E50M V1 and V8 for the following analysis.

### 3.2 Predictability and Downscaling

Next, we compare the spatial pattern of the forecast between NWPs and E50M 3D U-Net with both V1, which uses only the target variables, and V8, which includes all variables. The 3D U-Net model demonstrates significant improvements in both predictability and downscaling capabilities for temperature forecasts. While precipitation forecasts also show improvement, the gains are less pronounced than for temperature. For precipitation (Figs. 4 and S10), the 3D U-Net models achieve higher spatial resolution compared to NWP, revealing fine-scale patterns. However, a consistent underestimation of precipitation is observed across all lead times, with larger biases than those of the NWP model, particularly in coastal and mountainous regions, regardless of the number of input variables. Similar reductions in precipitation during downscaling and U-Net-based

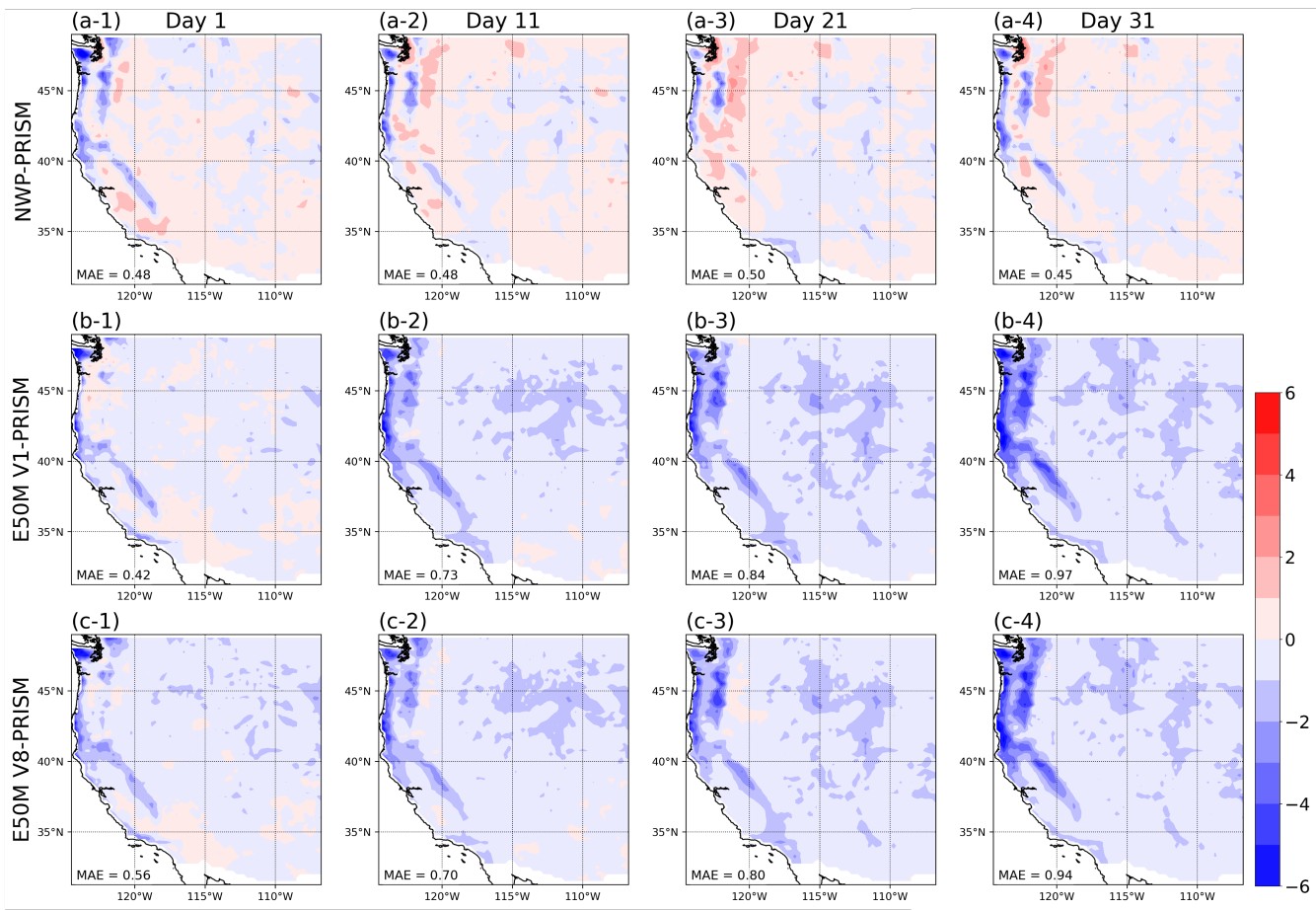

**Figure 4.** Comparison of precipitation forecasts (104 forecasts in 2023 are averaged) differences across lead times for the Western U.S. Rows represent (a) differences between NWP and PRISM, (b) differences between E50M V1 and PRISM, and (c) differences between E50M V8 and PRISM. Columns show forecasts for Days 1, 11, 21, and 31. Differences are depicted using the scale in the lower color bar (-6 to +6 mm/day).

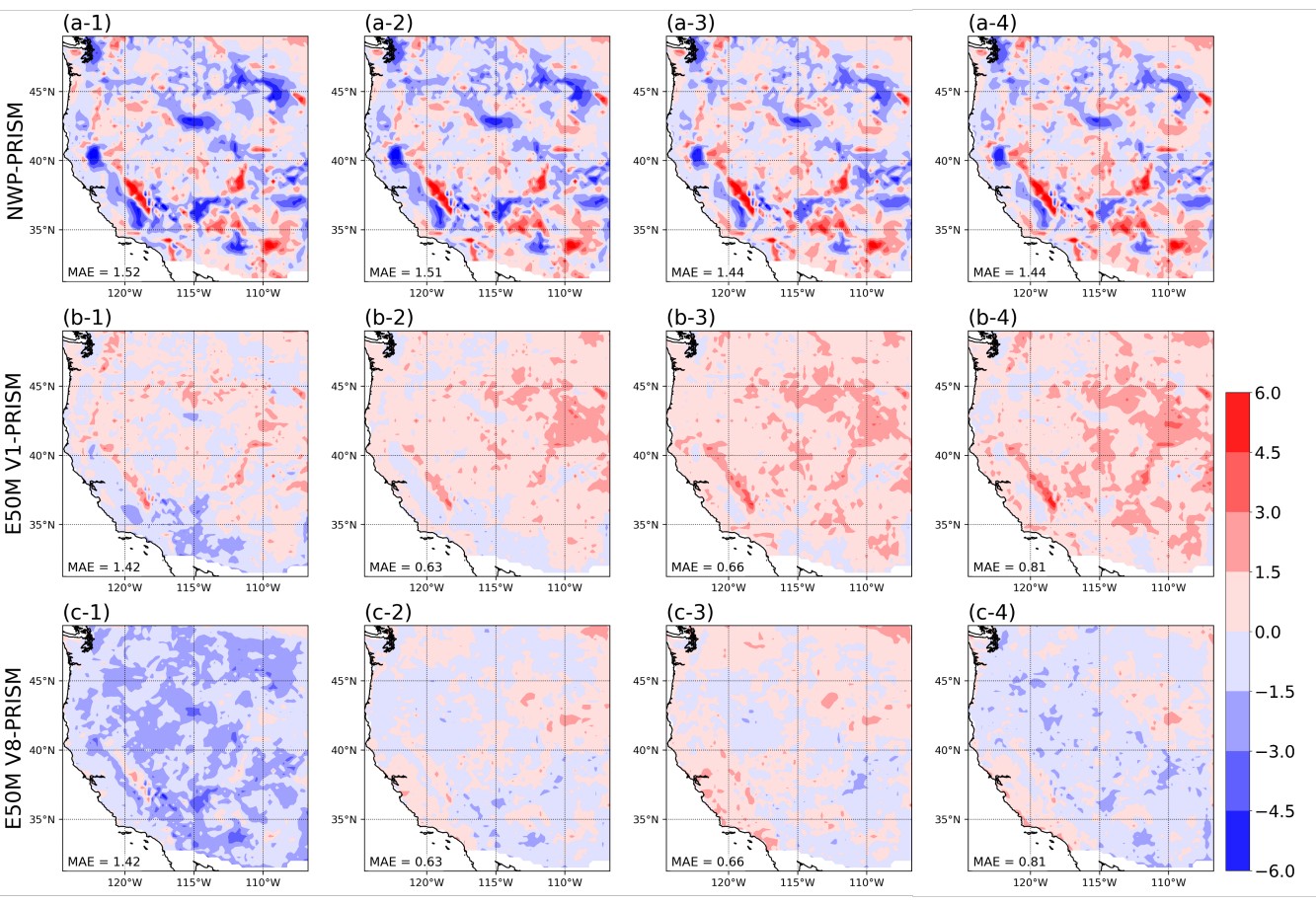

**Figure 5.** Temperature forecasts differences similar to Fig. 4. The layout is the same, with temperature differences shown in Kelvin (K).

post-processing have also been reported in other regions (Xin et al., 2024). Temperature forecasts (Figs. 5 and S11) showcase more substantial improvements. The 3D U-Net models significantly enhance spatial resolution and reduce overall forecast errors compared to NWP. The 3D U-Net approach, especially E50M V8, captures fine-scale temperature patterns effectively, showing reduced biases across various terrain types. Moreover, improvements were observed across the three dominant land cover types, which together account for over 80% of the study area (Figs. S7 and S8).

225

The performance of the 3D U-Net model in extreme cases provides further insights into its capabilities and limitations. Fig. 6 presents an extreme precipitation event in California from March 7 to March 13, 2023. The 3D U-Net models (E50M V1 and V8) demonstrate improved spatial detail compared to NWP. On March 10, the 3D U-Net model captures the rainfall that NWP doesn't (Fig. 6(b-4, c-4, d-4)) and specifies the location, both coastal area and inland, more accurately on 2023-03-11. Even so, the models still struggle with accurately capturing the intensity of heavy precipitation events. Increasing the training data can be one alternative to improve precipitation extremes (Hu et al., 2023). Alternatively, this limitation may stem from the post-processing technique itself and warrants further investigation.

230

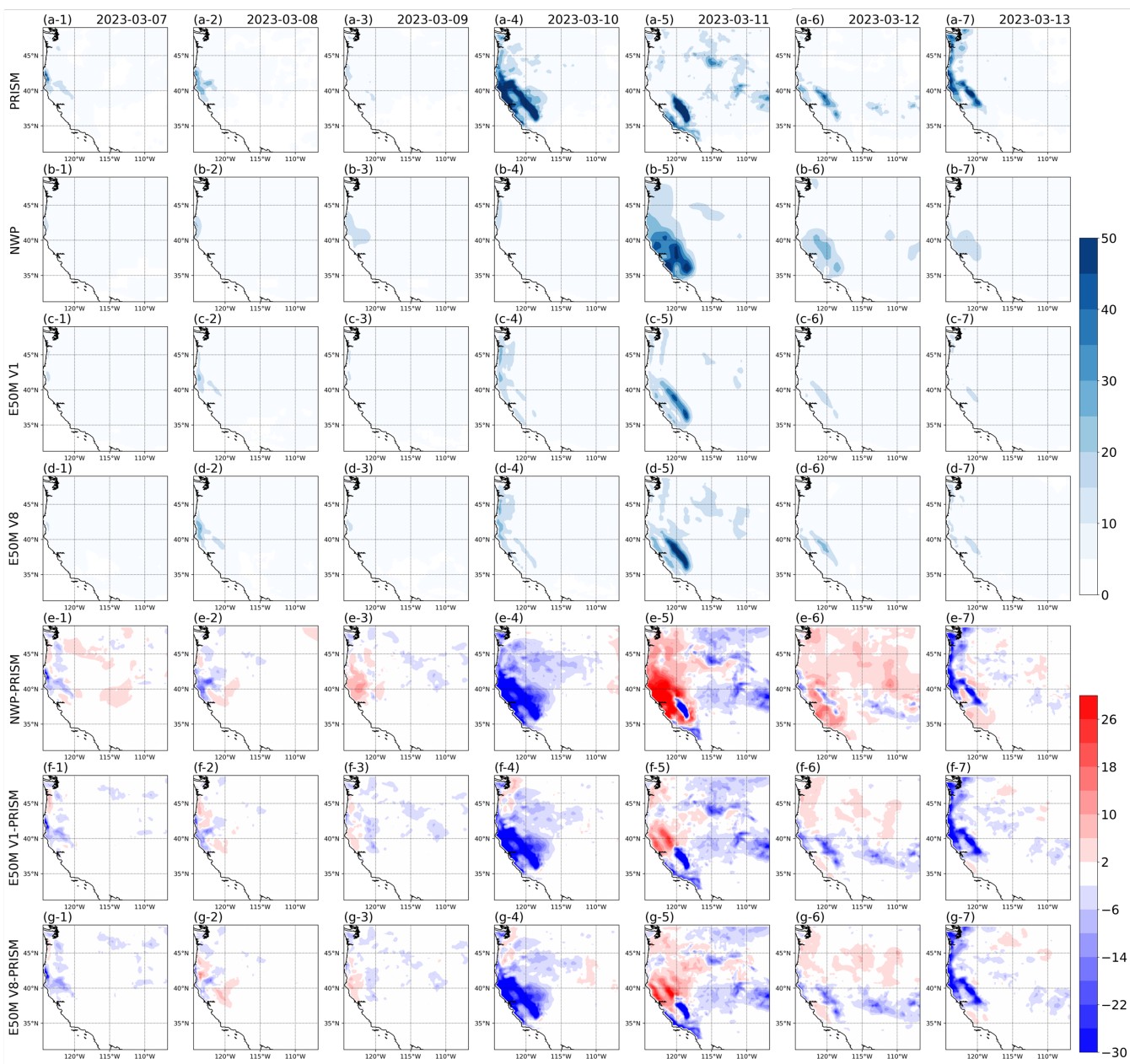

**Figure 6.** Daily precipitation forecasts for the Western U.S. from March 7 to March 13, 2023, with initial condition on March 6. In other words, March 7 (13) is the forecast with lead day 1 (7). Rows represent (a) PRISM observations, (b) NWP forecasts, (c) 3D U-Net E50M V1 predictions, (d) 3D U-Net E50M V8 predictions, (e) differences between NWP and PRISM, (f) differences between E50M V1 and PRISM, and (g) differences between E50M V8 and PRISM. Precipitation amounts (rows a-d) are shown using the scale in the upper color bar, while differences (rows e-g) are depicted using the scale in the lower color bar. Columns show forecasts for Days 1, 11, 21, and 31.

Even in extreme temperature cases, Figs. S12 and S13 confirm results that 3D U-Nets are superior to NWP. The overall performance of E50M V1 in the high-temperature case and E50M V8 in the low-temperature case, as well as the overall differences and recovery from cold waves, appear to outperform NWP. However, limitations are evident, highlighting the persistent challenges in predicting extreme events despite the improved spatial resolution.

The contrasting performance between precipitation and temperature forecasts underscores the varying complexities in predicting these two variables. Although some challenges are left in precipitation forecasting, the 3D U-Net model's ability to capture fine-scale patterns and improve spatial resolution for both variables represents a significant advancement. These results suggest that with further refinement, particularly in handling extreme events and complex terrain interactions, neural network-based post-processing methods like 3D U-Net have the potential to substantially improve both temperature and precipitation forecasts at sub-seasonal timescales.

## 3.3 Predictability in County-scale

To assess the model's performance at finer spatial scales, crucial for local decision-making and resource management, we evaluate forecasts for five selected county-level regions in the Western U.S. Fig. 7 presents comprehensive performance metrics for temperature and precipitation forecasts across these 5 regions (Fig. S1), comparing NWP with the most efficient model (E50M V1) over a 32-day lead time. Results for all models are shown in Fig. S14. The $E_{pre}$ metric was excluded for county-level results because its calculation requires spatial pattern correlation, which cannot be obtained from area-averaged values. For temperature forecasts at the county scale, 3D U-Net models generally demonstrate improved or comparable performance relative to NWP while the degree of enhancement varies significantly across regions. Along with the result of Section 3.1, E50M surpasses the other ensemble configuration's scores, and no conspicuous performance difference between varying the number of variables yields. Some areas, such as Seattle, show more pronounced enhancements in predictability, possibly due to the region's more uniform maritime climate. In contrast, areas with more complex terrain or microclimates show more modest improvements, highlighting the persistent challenges in downscaling to highly localized conditions. Incorporating land cover, which is already a key input in NWP models (López-Espinoza et al., 2020), could offer additional improvements in such regions.

For precipitation forecasts, 3D U-Net models enhance correlation on the weather scale but not on the sub-seasonal scale especially in two regions in Washington (Fig. 7(c-4,5)). Moreover, correlation exhibits higher variability in performance across different 3D U-Net configurations compared to temperature forecasts. Fig. 7(d) reveals a complex pattern. In most regions, 3D U-Net models and NWP show comparable RMSE values, with neither consistently outperforming the other across all lead times.

Note that the performance differences among 3D U-Net configurations for both targets are generally small at this county scale, while not identical to the patterns observed at larger spatial scales. This may be partly due to the very small size of the counties, which can increase uncertainty in the evaluation. As suggested by the land cover analysis, including a sufficiently large number of grids makes performance improvements more apparent, implying that the limited spatial coverage may have

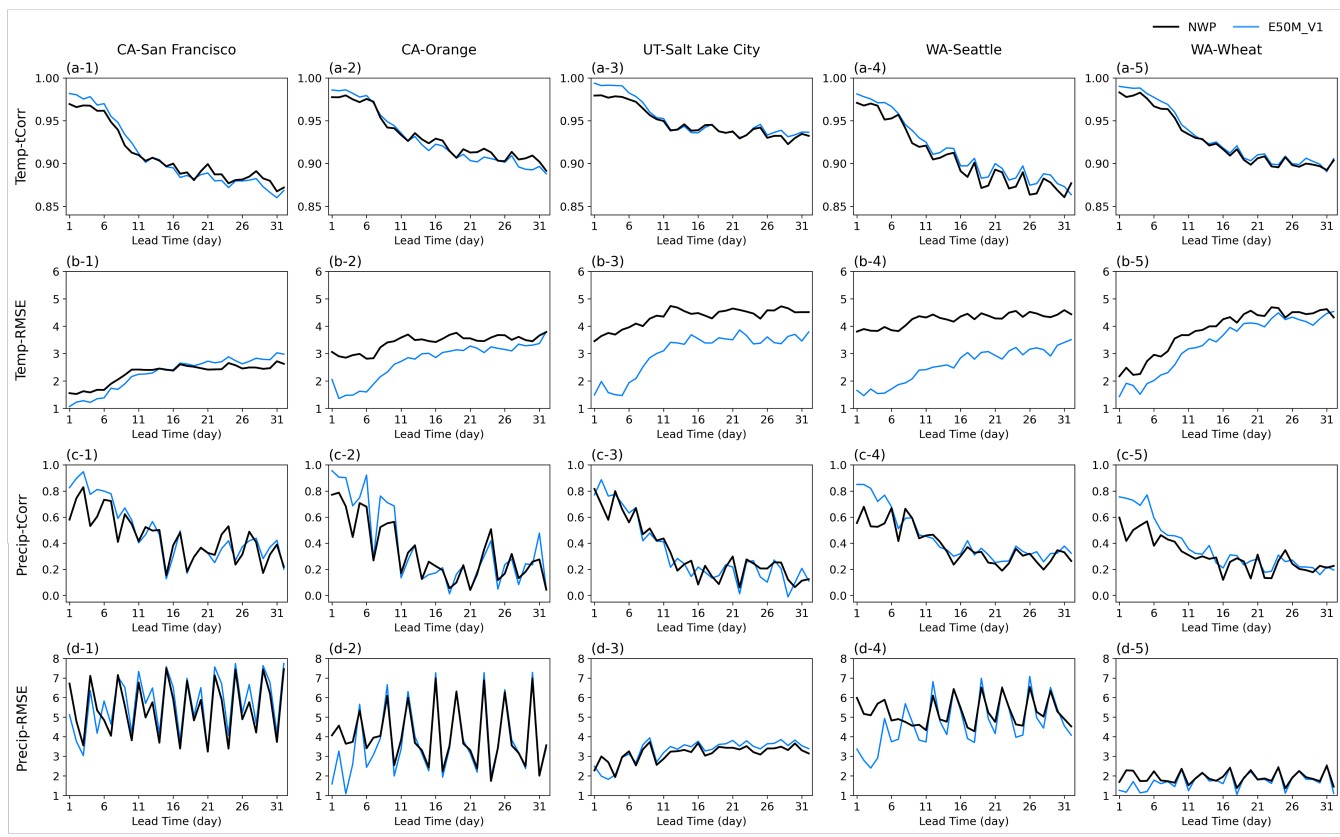

**Figure 7.** Performance metrics for temperature (Temp) and precipitation (Precip) forecasts across five county-level regions in the Western U.S., comparing NWP with the most efficient model (E50M V1) over a 32-day lead time. Metrics include $tCorr$ and RMSE. (a) Temperature correlation, (b) Temperature RMSE, (c) Precipitation correlation, (d) Precipitation RMSE. Columns represent different regions: (1) San Francisco, CA (2) Orange farm, CA, (3) Salt Lake City, UT, (4) Seattle, WA, and (5) Wheat farming area, WA.

constrained the observed benefits. Additionally, this implies that the benefits of increased model complexity may diminish at very fine spatial resolutions, where local factors become increasingly dominant.

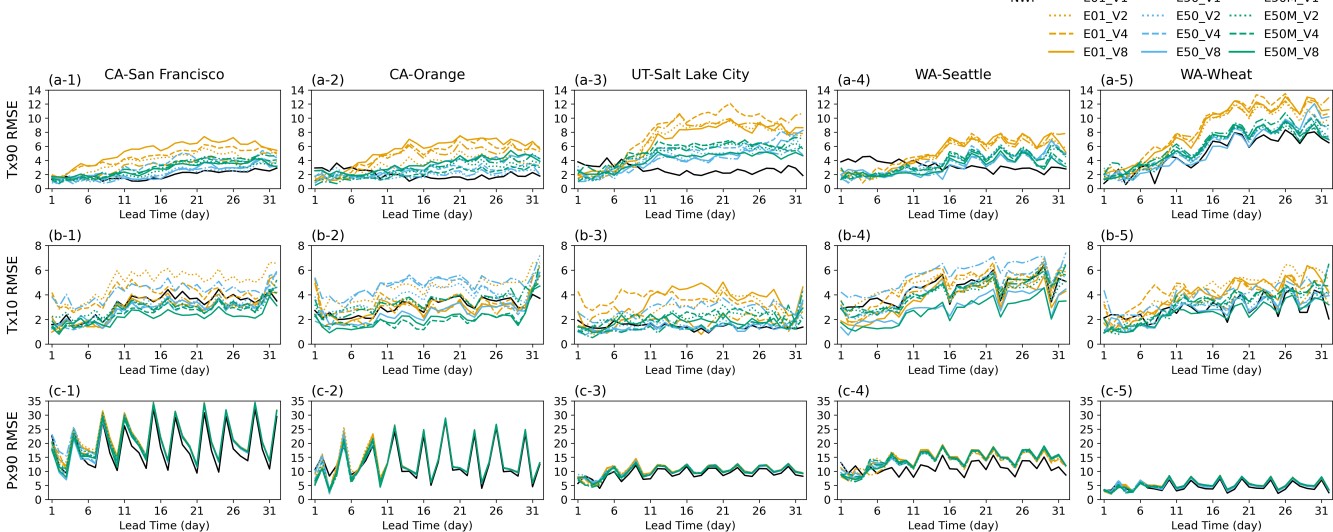

**Figure 8.** RMSE comparison across five U.S. counties for extreme temperature and precipitation forecasts. Results show NWP and various 3D U-Net configurations (E01, E50, E50M) with different input variables (V1-V8). Temperature metrics include 90th percentile (tx90) and 10th percentile (tx10). Precipitation uses 90th percentile (px90). (a) Temperature tx90 RMSE, (b) Temperature tx10 RMSE, (c) Precipitation px90 RMSE. Columns represent the same regions as in Fig. 7.

Fig. 8 elaborates the RMSE of each five regions regarding heat, cold, and precipitation extremes. Extreme events are defined as the 39 cases corresponding to the top 10% and bottom 10% of daily temperature, and the top 10% of daily precipitation, within the period from January 2023 to January 2024. The temperature and precipitation distributions for this period are shown in Fig. S15. Even though predicting heat extremes in Salt Lake City and Seattle in weather-scale is improved (Fig. 8(a-3,4)), 3D U-Net models don't outperform NWP in these extreme cases. Still, performance varies considerably on location, period, and extreme type. These can be attributed to several factors: Deep learning models are predisposed to yield results in which extreme values are smoothed out, called blurring effect (Lam et al., 2023), and tend to converge to the mean state (Bonavita, 2024). Furthermore, Olivetti and Messori (2024) highlights the similar result of Fig. 8 that global scale deep learning models often struggle with capturing the full range of variability in extreme events, especially in long-term prediction.

## 4    Conclusions

The findings of this study highlight the dual benefits of using the 3D U-Net architecture for sub-seasonal forecasting, namely enhanced accuracy and improved spatial resolution. By applying 3D U-Net-based post-processing to NWP models, the study demonstrated significant improvements in predicting both temperature and precipitation, especially in complex terrains and localized regions. The model's ability to downscale forecasts to higher spatial resolutions provided finer details, which are crucial for decision-making in regional disaster management. Furthermore, our results suggest that incorporating additional model-derived predictors or individual ensemble members yields limited improvement in sub-seasonal forecast postprocessing. Notably, the ensemble mean alone performs comparably to using the full set of ensemble components, pointing to a more computationally efficient alternative. These findings extend prior conclusions drawn from short-range forecasting studies (Rasp and Lerch, 2018; Schulz and Lerch, 2022; Höhlein et al., 2024) into the sub-seasonal prediction regime. Overall, the most efficient model was the ensemble average using only the target variables (E50M V1), and improvements were confirmed across all evaluation metrics except for the $E_{pre}$ index for precipitation. In particular, at a 32-day lead time, temperature and precipitation showed increases of 0.12 and 0.18, respectively, in the pattern correlation coefficient compared to NWP, along with reductions of approximately 31% and 22% in RMSE.

Nonetheless, some possible drawbacks remain evident. First and foremost, there was a spatial pattern improvement in precipitation, but the underestimation of precipitation in coastal and mountainous areas persisted. The added diversity in data could not resolve these limitations. Second, predicting extreme precipitation events with high accuracy is a challenging task. While the 3D U-Net could capture general patterns and improve spatial details, it still struggled to fully enhance extreme forecasts' accuracy.

The 3D U-Net model showed mixed performance for both temperature and precipitation forecasts at the county level. While 3D U-Net outperformed NWP models in predicting temperature such as in Seattle, its performance in precipitation forecasting was less consistent. The model was able to enhance spatial resolution and predictability for temperature at finer scales but struggled to deliver comparable improvements for precipitation. While the 3D U-Net model is effective for downscaling

temperature forecasts at the county level, further refinement is needed to improve its ability to capture precipitation patterns, particularly in regions with complex weather dynamics.

In conclusion, 3D U-Net's integration into sub-seasonal forecasting models offers substantial improvements such as capturing fine-scale weather patterns over traditional NWPs while maintaining computational efficiency. This model's ability makes it a promising tool for a wide range of atmospheric science applications, from short-term weather to sub-seasonal predictions. To move beyond "artificial neural network improves NWP," we emphasize operational feasibility and application value: an ensemble-mean, target-only configuration reduces input channels from 400 to 1–2, lowering memory and latency by more than two orders of magnitude and enabling daily, high-resolution S2S post-processing on commodity GPUs for routine water, fire, and agricultural decision-support. The approach is robust across seasons and land-cover types, yet skill still degrades for heavy precipitation in complex terrain, addressing these extremes and optimizing the complexity skill balance are priorities. To meet these challenges, we propose advancing into the probabilistic domain.

*Code and data availability.* The ECMWF perturbed forecast can be downloaded from https://apps.ecmwf.int/datasets/data/s2s/levtype=sfc/type=cf/. The PRISM dataset can be downloaded from https://prism.oregonstate.edu/. The model code is archived on Zenodo(Ryu et al., 2025), and at https://zenodo.org/records/14776781

*Author contributions.* The study was conceptualized by Jihun Ryu, and Jin-Ho Yoon. Jihun Ryu has done the data analysis, visualization, and writing the original draft. Hisu Kim has done the data analysis and writing the original draft. Reviewing and editing the manuscript is done by Shih-Yu (Simon) Wang and Jin-Ho Yoon.

*Competing interests.* The authors declare no competing interests.

*Acknowledgements.* This research is funded by the National Research Foundation of Korea under RS-2025-02363044 and the Korean Meteorological Agency under the grant KMI2018-07010.

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
