# Peer review of "Increasing Resolution and Accuracy in Sub-Seasonal Forecasting through 3D U-Net: the Western US"

_EGUsphere, 2025_

## Author Comment (AC1)

**Response to Anonymous Referee #1 comments for manuscript**

**Summary**

The paper proposes and tests a method for downscaling sub-seasonal weather forecasts to improve accuracy and spatial resolution. The approach uses a neural network architecture (3D U-Net) that has previously been used for similar tasks. The forecasts are from the ECWMF ensemble forecast system and the high-resolution data are from PRISM. The method is applied in the Western US. The effect of different input data (ensemble members versus mean, different variable sets) are examined. The authors find that the neural network improves temperature predictions relative to the original NWP forecasts. Results for precipitation are also presented but the quality is more mixed.

We greatly appreciate the reviewer's constructive comments and editorial suggestions, which have considerably improved this manuscript.

**High Level Feedback**

The topic itself is interesting, and the results shown (particularly for temperature) seem promising. However, it is difficult to evaluate the method due to the omission of important details. Some of the missing methodological details can be deduced by reading the code, but they should be provided in the paper itself.

Thank you for your helpful comment. We have added additional U-Net details to the main text to improve clarity.
Line 108: The 3D U-Net model was trained for 100 epochs using the adam-optimizer with an initial learning rate of 1e-4 and a batch size of 11, selected based on GPU memory limitations (Kingma and Ba, 2017). The network architecture consists of three encoding and decoding blocks, each composed of 3D convolutional layers with $3\times3\times3$ kernels. Average pooling was used for downsampling in the encoder, and transposed convolution was used for upsampling in the decoder. The GeLU activation function was applied after each convolutional layer. To prevent overfitting, we applied early stopping based on validation loss with a patience of 10 epochs. The loss function combines mean squared error (MSE) and spatial pattern correlation, with equal weighting assigned to both components. We chose this combination because each metric emphasizes a different aspect of prediction performance. MSE evaluates the model's ability to reproduce the absolute magnitude of values, while spatial pattern correlation captures the fidelity of the overall spatial distribution, which is particularly important in sub-seasonal forecasting. All configurations were selected through trial-and-error experiments to ensure training stability and generalization capability. These details have been incorporated into the main manuscript for transparency.
In additional to technical omissions, a main concern is the lack of clarity around the purpose and impact of the analysis. The main question, as I understand it, is: what is the effect of using (a) different sets of predictor variables and (b) different ensemble components/aggregations on prediction accuracy? The introduction states that Höhlein et al. (2024) examined (b) and reached approximately the same conclusion as this study. How is this study different? Question (a) seems relevant. However, there is very little discussion of the specific predictor variables (I think they are only mentioned in the SI) and how/why specific variables may contribute to better or worse predictions. Again, it is not clearly explained how

this study differs from the cited Horat and Lerch (2024) or Weyn et al. (2021). Most of the framing of the results and conclusions boil down to "neural network downscaling improves prediction accuracy relative to NWP," which, based on the introduction, seems to already be well-established.

We appreciate this concern and now explicitly clarify our novelty in both scope and approach. Prior studies demonstrated diminishing returns from additional predictors or member-wise inputs in short-range ($\leq$ 120 h) post-processing (Höhlein et al., 2024; Rasp and Lerch, 2018; Schulz and Lerch, 2022). What was unknown is whether these findings extend to sub-seasonal (up to 32 days) daily targets, where regime transitions (*e.g.*, MJO) and aggregation effects alter skill decay. Our 3D U-Net treats lead time as a learnable dimension, enabling transfer of shorter-lead skill to longer leads at daily resolution over complex U.S. terrain.

In addition, we have reframed the impact statement to go beyond "NN improves NWP" by emphasizing operational feasibility and application value. The ensemble-mean, target-only configuration reduces input channels from 400 to 1–2, lowering memory and latency demands by over two orders of magnitude. This enables daily, high-resolution S2S post-processing on commodity GPUs—making the approach viable for routine updates in water, fire, and agricultural decision-support systems. We also highlight robustness to season and land cover type, and candidly note where skill still degrades (*e.g.*, heavy precipitation in complex terrain), which we link to the probabilistic roadmap in the following response.

We have added a concise comparison table to the Introduction contrasting our inputs, horizon, architecture, and findings with cited works, and framed our hypotheses explicitly:

H1: 3D U-Net with lead-time convolutions yields daily S2S skill gains over raw NWP.

H2: At S2S leads, ensemble-mean is equivalent to all-members for deterministic targets.

H3: Auxiliary predictors beyond the target add minimal incremental skill.

This framing and table now appear in the revised manuscript to sharpen the "what's new" message.

Line 42: Prior evaluations of predictor sets and ensemble usage have largely been limited to short lead times ($\leq$5 days) and single valid times (Rasp and Lerch, 2018; Schulz and Lerch, 2022; Höhlein et al., 2024), probing predictability at an instant (Table 1). In contrast, we target sub-seasonal forecasting by supplying sequences of forecast lead times to encode, thereby extending previous findings to lead times longer.

This study enhances predictability in the Western United States through the 3D U-Net-based post-processing that encodes temporal information via forecast lead times and downscaling forecasts to higher spatial resolutions.

Line 283: Furthermore, our results suggest that incorporating additional model-derived predictors or individual ensemble members yields limited improvement in sub-seasonal forecast postprocessing. Notably, the ensemble mean alone performs comparably to using the full set of ensemble components, pointing to a more computationally efficient alternative. These findings extend prior conclusions drawn from short-range forecasting studies (Rasp and Lerch, 2018; Schulz and Lerch, 2022; Höhlein et al., 2024) into the sub-seasonal prediction regime.

Line 306: To move beyond "artificial neural network improves NWP," we emphasize operational feasibility and application value: an ensemble-mean, target-only configuration reduces input channels from 400 to 1–2, lowering memory and latency by more than two orders of magnitude and enabling daily, high-resolution S2S post-processing on commodity GPUs for

routine water, fire, and agricultural decision-support. The approach is robust across seasons and land-cover types, yet skill still degrades for heavy precipitation in complex terrain, addressing these extremes and optimizing the complexity skill balance are priorities. To meet these challenges, we propose advancing into the probabilistic domain.

Table 1: Comparison of the proposed method with previous post-processing studies.

| Study | Type | Input | | Output | |
|---|---|---|---|---|---|
| | | Ensemble | Variable | Lead time | Model by lead time |
| Our study | post-processing | individual member | used additional variables | 0 – 32 days, daily | One model for forecast period |
| Rasp and Lerch, 2018 | post-processing | mean, std | used additional variables | 48h | Lead time specific model |
| Schulz and Lerch, 2022 | post-processing | mean, std, individual member | used additional variables | 0–21 h, hourly | Lead time specific model |
| Höhlein et al., 2024 | post-processing | individual member | used additional variables | wind gust: 6h, 12h, 18h temperature: 24h, 72h, 120h | Lead time specific model |
| Horat and Lerch, 2024 | post-processing | mean | used additional variables | temperature: 3–4W, 5–6 W mean precipitation: 3–4W, 5–6 W accumulate | Lead time specific model |

The motivation for the ensemble-based predictors is also confusing. The purpose of an ensemble prediction system is to represent uncertainty, which is not discussed. Also the ensemble members are simulations that, by construction, do not start from the "optimal" estimate of the initial conditions. So it is not surprising that E01 performs worse (unless by "first" ensemble member you mean the control). Interpreting the relative performance of E50 versus E50M requires methodological details that are not provided. But again it is not surprising that the performance is similar given that E50 output is being reduced to a deterministic prediction. It seems like the value of downscaling based on an ensemble would be more in representing forecast uncertainty than improving deterministic downscaled predictions

We agree that the primary added value of an ensemble lies in uncertainty representation. In our deterministic framework, member-wise inputs did not outperform the ensemble mean, suggesting limited exploitation of ensemble spread. We have clarified this in the Discussion and added a short "Future Work" paragraph noting how this could be addressed:

1. Training probabilistic versions of the 3D U-Net (*e.g.*, quantile regression with pinball loss, CRPS-optimized Gaussian output layers; Hersbach 2000; Gneiting & Raftery 2007).

2. Evaluating reliability and spread–skill using Brier scores, rank histograms, and calibration methods (isotonic regression).

3. Testing Bayesian or ensemble-based deep learning methods to better utilize ensemble spread.

While outside the current scope, these extensions are feasible and would align the model more closely with the ensemble's intended purpose. Also, we have also added more detailed methodological descriptions in the manuscript to clarify how the ensemble configurations were used.
Line 191: Our current approach produces deterministic forecasts and therefore cannot fully represent the uncertainty that NWP ensembles are designed to capture. To address this limitation, future work could consider several directions. One option is to train the 3D U-Net to generate probabilistic forecasts, for example via quantile regression with a pinball loss or by predicting parametric distributions (*e.g.*, Gaussian) optimized with the Continuous Ranked Probability Score (CRPS) (Hersbach, 2000; Gneiting and Raftery, 2007). Another is to evaluate the reliability and the relationship between spread and skill using Brier scores, rank

histograms, and calibration methods such as isotonic regression. Finally, more advanced avenues could include modifying the network to produce its own ensemble or adopting Bayesian deep learning frameworks. Line 105: For example, the E50 V8 configuration has 400 input channels, while the E50M V2 configuration has 2 input channels. In the model, the input dimensions are referred to as height, width, and depth, corresponding to lead time, latitude, and longitude, with sizes of 32, 72, and 72, respectively.

**Specific Feedback**

1. Either the $E_{pre}$ formula or the subsequent description of it is incorrect. You say $E_{pre} = 0$ for perfect predictions, which would require the term inside the square brackets to equal 1. However, it is $2^2 = 4$ when $sigma_{obs} = sigma_{pre}$ and $r_0 = r_i = 1$. It would also be helpful, if possible, to provide some intuition for the terms in this statistic. E.g., why is the standard deviation term squared and the correlation terms raised to the fourth?

   Thank you for pointing that out. There was an error in the original formula, which has now been corrected. This metric was proposed by Yang et al. (2013), and we have followed their method in our study. According to their explanation, certain observational datasets exhibit high spatial pattern reliability, and therefore the correlation term was raised to the fourth power to give it greater weight in the evaluation.

$$E_{pre} = \frac{1}{N} \sum_{i=1}^{N} \log \left[ \frac{\left( \frac{\sigma_{obs,i}}{\sigma_{pre,i}} + \frac{\sigma_{pre,i}}{\sigma_{obs,i}} \right)^2 (1 + r_0)^4}{4 (1 + r_i)^4} \right] \tag{1}$$

2. Regarding skip connections, for a given level (or spatial resolution) in the u-net, shouldn't there be twice the number of channels in the first layer on the right side of the U as on the last layer on the left (due to the concatenation of feature maps)? This is what is shown in both the Horat and Ronneberger papers. Also, I couldn't determine where the skip connections were implemented in the code but maybe I just missed it.

3. Related to (2), there is no description of the convolution operations (e.g., kernel size). The pooling operation is also not in the body of the text, only in Fig 1 (but not defined). These are scientifically important details considering they control the spatial scales at which information can be extracted.

4. The activation function(s) is also not stated.

5. The loss function is mentioned but not explicitly stated. How are relative weights of the correlation and MSE terms set? Also, point/cell-wise MSE and correlation are closely related so what is the value of including both terms?

   Thank you for the detailed and thoughtful comments. Because comments 2–5 are closely related, we address them together here. As the reviewer correctly noted, the number of channels in the decoder should double at each stage due to concatenation with the corresponding skip-connection features. Our experiments followed this structure. The original schematic may have been confusing because it showed only the post-convolution

channel counts at each layer rather than the doubling that occurs immediately after concatenation. We have revised Figure 1 to clarify the architecture: subscripts beneath each tensor block denote the channel counts, and the operations at each step are indicated by arrows within a dashed box with brief descriptions. We have also added the requested model details to the main text to improve clarity and reproducibility.

The loss function combines MSE and spatial pattern correlation, with equal weighting assigned to both components. The loss function combines mean squared error (MSE) and spatial pattern correlation, with equal weighting assigned to both components. These two metrics capture different aspects of model performance: pattern correlation evaluates how well the model reproduces the spatial characteristics of a region, while MSE measures the absolute error in magnitude. We therefore include both terms in the loss, and this explanation has also been added to the main text.

Line 108: The 3D U-Net model was trained for 100 epochs using the adam-optimizer with an initial learning rate of 1e-4 and a batch size of 11, selected based on GPU memory limitations (Kingma and Ba, 2017). The network architecture consists of three encoding and decoding blocks, each composed of 3D convolutional layers with $3\times3\times3$ kernels. Average pooling was used for downsampling in the encoder, and transposed convolution was used for upsampling in the decoder. The GeLU activation function was applied after each convolutional layer. To prevent overfitting, we applied early stopping based on validation loss with a patience of 10 epochs. The loss function combines mean squared error (MSE) and spatial pattern correlation, with equal weighting assigned to both components. We chose this combination because each metric emphasizes a different aspect of prediction performance. MSE evaluates the model's ability to reproduce the absolute magnitude of values, while spatial pattern correlation captures the fidelity of the overall spatial distribution, which is particularly important in sub-seasonal forecasting. All configurations were selected through trial-and-error experiments to ensure training stability and generalization capability. These details have been incorporated into the main manuscript for transparency.

[Figure]

Figure R1: (Figure 1) Schematic of the 3D U-Net architecture adapted for weather forecast post-processing. The model consists of a contracting path (left, green), an expanding path (right, blue), and a bottleneck layer (center, teal), with skip connections (dashed gray arrows) preserving spatial information. Operations between layers are described in the dashed box on the right corner.

6. The claim that pattern correlation and RMSE "quantify the model's ability to capture the spatial patterns" is not justified. Grid cell-wise RMSE is invariant under spatial permutation. The formula or weighting scheme for pattern correlation is not given. From the code it looks like weights are assigned inversely to latitude?

Thank you for pointing this out. We agree that RMSE is not sensitive to spatial structure, as it is computed pointwise and is invariant under spatial permutation. In contrast, pattern correlation captures the agreement in spatial distribution between the prediction and the observation, and is more sensitive to spatial structure. To clarify, we have revised the manuscript to explicitly distinguish the roles of the two metrics. Also, upon rechecking our implementation, we confirm that pattern correlation is computed using cosine latitude weighting (i.e., proportional to the area of each grid cell), not the inverse of latitude.
Line 147: Pattern correlation evaluates the model's ability to reproduce the spatial distribution of temperature and precipitation fields, while RMSE quantifies the average magnitude of forecast errors at each grid point.

7. The accuracy assessment would benefit from disaggregation into bias versus "random" errors. Using unbiased RMSE is a way of doing this (see e.g., Entekhabi et al., 2010). In figures 4 and 5, it looks like the downscaled predictions have meaningful biases for certain combinations of variable sets and time steps. For example, panel g4 in Fig. 4 and panel g1 in Fig. 5. If it turns out that the predictions are not "on average" biased, the disaggregation might be less important. However, the presence/absence of bias overall should be mentioned given the visually apparent biases in the figures. This analysis would help identify how much of model performance is coming from downscaling versus simply bias correcting the ECMWF forecasts.

Thank you for your constructive comment. As shown in the Fig. R2, we calculated RMSE separately for bias and unbiased RMSE. While the NWP forecasts exhibited systematic biases, the deep learning models also showed biases to some extent. In particular, for precipitation, the deep learning models demonstrated a persistent underestimation bias compared to NWP, which we briefly mentioned in Line 164. Nevertheless, we found that the deep learning models still reduced the unbiased RMSE relative to NWP, indicating an improvement in predictive skill beyond bias correction alone. We have revised the manuscript to clarify this point.
Line 219: However, a consistent underestimation of precipitation is observed across all lead times, with larger biases than those of the NWP model, particularly in coastal and mountainous regions, regardless of the number of input variables. Similar reductions in precipitation during downscaling and U-Net-based post-processing have also been reported in other regions (Xin et al.,2024).

[Figure]

Figure R2: Scores comparing NWP and 3D U-Net models (E01, E50, E50M) for temperature (top row) and precipitation (bottom row) forecasts over 32 days. Columns show (a, d) RMSE, (b, e) bias, and (c, f) unbiased RMSE, respectively

8. The possibility of systematic errors (e.g., season-dependent bias) should be acknowledged/discussed. Even if the predictions are not "on average" biased, there may still be systematic errors. Given that the test period spans one full year, seasonal biases could cancel out such that the predictions appear unbiased in aggregate. The NN predictions may tend toward the overall mean of the training data leading to seasonal biases. I.e., the NN could be introducing systematic biases not present in the NWP output. This may or may not be happing but seems important to consider and rule out.

Thank you for your feedback. We presented our results using forecast data for 104 initialization dates in 2023. We analyzed the evaluation indices for DJF, MAM, JJA, and SON based on the initialization dates and present these results in Figs R3 and R4. The results show improvements in all performance indices, except for precipitation $E_{pre}$, across all seasons. For precipitation $E_{pre}$, performance decreased in spring and summer but increased in winter and spring. Based on these results, we conclude that seasonal bias is minimal. We have added these figures to the supplementary materials and included a discussion of these findings in the main text.

Line 167: Before conducting a detailed analysis of the results, we examined the potential for seasonal bias and the performance by land cover type. Our findings show improvements in all seasonal evaluation metrics for both temperature and precipitation, except for precipitation $E_{pre}$ in spring and summer (Figs.S4 and S5). This suggests that the enhanced performance is not simply due to the model converging toward the seasonal mean across all seasons. Rather, the improvements reflect the model's ability to capture relevant patterns within each season.

[Figure]

Figure R3: (Figure S4) Seasonal benchmark scores for the Western U.S., comparing NWP and 3D U-Net models for precipitation forecasts. The scores are categorized by initialization dates for DJF (first row), MAM (second row), JJA (third row), and SON (fourth row).

[Figure]

Figure R4: Figure S5) Same as Fig. R3, but for temperature forecasts.

9. I don't see the train/test procedure for E50 anywhere. There are several ways the training and testing could work, so it should be specified. This detail is essential given that this seems to be one of the main focuses of the paper.

Thank you for your feedback. We have clarified the training and testing procedure in the revised manuscript. Specifically, ensemble members and variables are combined along the channel dimension. For example, in the experiment using all 50 ensemble members and 4 variables (E50 V4), the model receives 400 input channels. We have added this explanation and an illustrative example to the manuscript for clarity. All other training settings, including data splitting, model architecture, and optimization procedures, are kept consistent across E01, E50, and E50M configurations to ensure a fair comparison. Line 105: For example, the E50 V8 configuration has 400 input channels, while the E50M V2 configuration has 2 input channels. In the model, the input dimensions are referred to as height, width, and depth, corresponding to lead time, latitude, and longitude, with sizes of 32, 72, and 72, respectively.

10. I don't think the predictor variable sets are ever actually stated. The top 8 predictors for each target variable are given in the appendix, but this is scientifically important information that should be in the main text. It is also not clear from the text or SI which of the 8 are included in each subset (I see from the code that Vk is the k highest correlation with the target, which makes sense but is not stated).

Thank you for constructive comment. We have moved the Table S1 from the appendix to the main text. Additionally, we have added a sentence clarifying that the $Vk$ subset represents the top $k$ predictors selected based on their spatial correlation with the target variable.
Line 133: The spatial pattern correlation coefficient between the mean state of each additional variable and that of the target variable is then computed. The absolute values of these correlations are averaged across the two timescales, and the variables are ranked accordingly. Rankings are shown above each bar in Fig.S2. The top eight variables for each target are selected for use in the 3D U-Net model, as summarized in Table 2.

11. I think it is important to acknowledge somewhere in the paper that PRISM data are the output of statistical interpolation and so have uncertainties and systematic errors, which will impact the output of the NN. The paper sometimes refers to PRISM data as "observations," which is misleading.

We agree with your point and have revised the manuscript accordingly. We have replaced references to PRISM data as "observations" with more accurate terminology and now refer to them as a "reanalysis dataset".

**Minor Comments**

Line 16 (and elsewhere): when you refer to a paper in running text, you should still provide the year in parentheses

Thank you for the suggestion. We have revised the in-text citations throughout the manuscript to include the publication year in parentheses when referring to papers in running text.

Line 34: Re "subset of variables," a subset of what? I think you're referring to "additional" or maybe "auxiliary" variables, but I don't think subset is the right term. If anything it's a (super)set that includes the target variables as a subset.

We agree with the reviewer that the term "subset" was imprecise. We have revised the text to clarify that we are referring to "set of additional variables". // Line 37: a broader set of additional variables

Line 36: Again I do not think "sub-variable" is the right term here. See above for suggested alternatives.

Thank you for pointing this out. We have replaced "sub-variable" with "additional variable" to maintain consistency and improve clarity.

Lines 49-51: On line 49 is says "we select forecasts from CY40R1" and on line 51 it says "We utilize forecasts from CY40R1 to CY48R1." This might be clearer to someone more familiar with ECMWF forecast naming scheme, but I find this confusing. It would be helpful to give some additional explanation what this means and maybe provide a link to the relevant data product(s).

We appreciate this comment and have revised the text to clarify that we ultimately used forecasts dataset for our experiments. We have also added a link to the ECMWF forecast model documentation to help readers unfamiliar with these cycle names. Line 57: We select the $1.5° \times 1.5°$ resolution, 50 ensemble perturbation forecasts (approximately 120 km $\times$ 120 km over the study region), twice-weekly forecast cycles, and 32-day lead times to match the earliest version of the ECMWF model (Roberts et al., 2018).
Line 61: For detailed information on each version of the model, please refer to the ECMWF model archive: https://confluence.ecmwf.int/display/S2S/ECMWF+Model.

Line 70: Remove "properly." Also, I believe the preferred GMD style is "Fig." rather than "Figure" in running text.

Thank you for feedback. "Properly" has been removed for clarity. We have also changed "Figure" to "Fig." to align with GMD style guidelines.

Line 73: I generally find it clearer to talk in terms of fine versus coarse spatial resolution.

Thank you for your helpful suggestion. In response, we have revised the description of the 3D U-Net structure to use the terms "fine" and "coarse" spatial resolution, which we agree provide greater clarity.
Line 89: The contracting path progressively reduces spatial dimensions (moving from fine to coarse) while increasing feature channels, allowing the model to capture broader contextual information. Conversely, the expanding path restores spatial resolution (from coarse to fine), enabling precise localization of weather patterns.

Line 96: Regarding "conservative interpolation," can you be more specific about the method?

Thank you for your question. We have clarified the conservative interpolation in text. This method preserves the integrated quantity over the grid area, which is particularly important for variables like precipitation.
Line 128: We then apply conservative interpolation, a method that preserves physical quantities like mass or energy during spatial grid adjustments, to ensure the accurate preservation of values during spatial adjustments.

Lines 97-98: Regarding "given the established relationship ...", what is the relationship?

Thank you for constructive comment. Here, "established relationship" refers to the finding that predictability can be evaluated based on the mean state, as described by Ryu et al. (2024). This sentence has been partially revised for clarity.

Line 131: Based on the fact that predictability can be evaluated using the mean state (Ryu et al., 2024), we calculate the mean state of each additional variable across both weather and sub-seasonal timescales.

Line 110: semicolon should be colon

Thank you for feedback. Corrected the semicolon to a colon as suggested.

First paragraph of 3.1: This description of the scope of analysis should come much earlier in the paper, not in the results section.

We appreciate the suggestion. However, this paragraph is intended as a brief summary of the experimental settings used for analysis, not as a statement of the overall study scope. To clarify, we have revised the paragraph accordingly.
Line 161: The performance of the 3D U-Net model, compared to traditional NWP forecasts, was evaluated across twelve cases combining three ensemble configurations and four input variable sets (Fig.S3). The 3D U-Net consistently outperformed the raw NWP forecasts across three evaluation metrics, except for $E_{pre}$ in precipitation. Statistical tests comparing each model's evaluation metrics with those of the NWP baseline showed that, apart from the $E_{pre}$ metric for precipitation, the improvements were significant. For precipitation $E_{pre}$, the results were mixed: five models (E01 V4, E50 V2, E50 V8, E50M V1, and E50M V2) showed no significant improvement, while seven models exhibited significant degradation.

Line 131: I think you mean "metrics"

Thank you for feedback. Corrected "matrices" to "metrics."

Line 141: "suggest elevation can enhance temperature post-processing accuracy" this is just lapse rate, right? I don't think this should be framed as a finding of NN methods

Thank you for pointing this out. We agree with the reviewer that the positive impact of elevation on temperature post-processing accuracy primarily reflects the lapse rate, and thus should not be considered a novel finding of neural network methods. However, incorporating elevation enables the model to better learn and correct for this effect, thereby improving performance. This result aligns with previous studies, such as Rasp and Lerch (2018), which identified elevation as the most important predictor for temperature post-processing. This sentence has been partially revised for clarity.
Line 200: This may be attributed to the inclusion of altitude, which has been shown to be one of the most important variables in temperature post-processing (Rasp and Lerch, 2018).

**References**

Höhlein, K., Schulz, B., Westermann, R., & Lerch, S. (2024). Postprocessing of ensemble weather forecasts using permutation-invariant neural networks. *Artificial Intelligence for the Earth Systems*, *3*(1), e230070.

Horat, N., & Lerch, S. (2024). Deep learning for postprocessing global probabilistic forecasts on subseasonal time scales. *Monthly Weather Review*, *152*(3), 667–687.

Rasp, S., & Lerch, S. (2018). Neural networks for postprocessing ensemble weather forecasts. *Monthly Weather Review, 146*(11), 3885–3900.

Schulz, B., & Lerch, S. (2022). Machine learning methods for postprocessing ensemble forecasts of wind gusts: A systematic comparison. *Monthly Weather Review, 150*(1), 235–257.

---

## Author Comment (AC2)

**Response to Anonymous Referee #2 comments for manuscript**

**Summary**

This manuscript revolves around a deep learning-based method for postprocessing subseasonal weather forecasts in the Western United States. The authors used multiple 3D U-Net architectures. These models use ensemble forecasts from the ECMWF model as input. As the target variables, the high-resolution temperature and precipitation data from the PRISM dataset are used. The authors investigate how model performance varies with using only one member of the ensemble, the full ensemble, and the mean of the ensemble. The authors also explore the impact of the number of input variables from the ECMWF forecasts. The performance of the models is evaluated using RMSE, pattern correlation, and the E_pre index. The analysis focuses on both regional (full-domain) and county-level scales, covering urban and agricultural areas. As a final analysis, the authors also examine the performance of the models for extreme weather events.

Results indicate that the 3D U-Net models improve temperature forecasts compared to the raw ECMWF outputs. The models enhance spatial resolution for precipitation but tend to underestimate intensity, particularly in coastal and mountainous regions. Increasing the number of input variables or using all 50 ensemble members shows little to no gain in performance. While the model offers clear benefits at larger spatial scales, these advantages are smaller at finer resolutions and for extreme weather conditions, where performance becomes more comparable to the baseline.

We greatly appreciate the reviewer's constructive comments and editorial suggestions, which have considerably improved this manuscript.

**Overall assessment and recommendations:**

The proposed method for postprocessing the ECMWF weather forecast is well thought out and novel. The authors have invested significant effort into optimizing the models. This has been done by considering the number of sub-variables used from the forecast as input features for the U-net models and the number of ensemble members to include. It's also good that the analysis also includes evaluations on specific regions (counties) with varying climates. Here, the U-Net models' results are a bit lackluster, but this can be further investigated in a follow-up paper. The additional investigation into extreme events is valuable, as it pushes the model to its limits and offers insight into its performance for less frequent but very important weather conditions. The manuscript is well written and follows a logical structure. There are some minor typographical errors, most of which are listed below.

Overall, the obtained results and the paper itself are definitely promising, but they could be further improved with some additional explanation. The main shortcoming is the lack of a more thorough justification for certain choices made throughout the analysis. I recommend expanding the main text and supporting information to justify and contextualize these decisions better.

Thank you for your constructive feedback. We have addressed all of your comments by including more detailed information in the main text.

**General comments**

For me, the authors don't explain the reasoning behind the choice of model clearly enough. Using a U-Net makes sense for this problem, but the 3D aspect isn't explained or justified well enough. Adding another paragraph of explanation in Section 2.2 could already go a long way here. Right now, it's not clear why a 3D U-Net was used instead of a more standard 2D version with lead time added as a channel, for example. The 3D architecture might be the better option, but the manuscript should explain how and why that decision was made more clearly.

Thank you for pointing this out. We used a 3D U-Net architecture because many climate variables, such as temperature and geopotential height, exhibit temporal continuity. Specifically, we treated lead time as the height dimension, based on the assumption that shorter lead-time forecasts—generally more accurate which are generally more accurate time predictions, such as those on a sub-seasonal timescale. Additionally, the 3D formulation allows for daily-scale resolution in sub-seasonal forecasts, rather than relying on weekly-mean aggregation. This explanation has been added to Section 2.2, as suggested by the reviewer.

Line 81: We add the height dimension to implement the 3D U-Net structure, accounting for the temporal continuity inherent in meteorological variables such as temperature and geopotential height, as shown in Fig. 1. In this framework, lead time is treated as the vertical dimension. This allows the model to utilize information from shorter lead times, which typically exhibit higher predictive skill, to improve sub-seasonal forecasts. Additionally, this structure enables the generation of daily forecasts, in contrast to traditional approaches that rely on weekly averages, thereby providing a finer temporal resolution for downstream applications.

Additionally, in the main section or the supporting information, the authors should include more details on the chosen model configurations, parameters, and hyperparameters. Details such as the number of epochs, batch size, kernel size, and methods to prevent overfitting aren't documented. Making all these elements as transparent as possible is crucial for a machine learning paper.

Thank you for your constructive comment. We have added additional U-Net details to the main text to improve clarity.

Line 108: The 3D U-Net model was trained for 100 epochs using the adam-optimizer with an initial learning rate of 1e-4 and a batch size of 11, selected based on GPU memory limitations (Kingma and Ba, 2017). The network architecture consists of three encoding and decoding blocks, each composed of 3D convolutional layers with $3\times3\times3$ kernels. Average pooling was used for downsampling in the encoder, and transposed convolution was used for upsampling in the decoder. The GeLU activation function was applied after each convolutional layer. To prevent overfitting, we applied early stopping based on validation loss with a patience of 10 epochs. The loss function combines mean squared error (MSE) and spatial pattern correlation, with equal weighting assigned to both components. We chose this combination because each metric emphasizes a different aspect of prediction performance. MSE evaluates the model's ability to reproduce the absolute magnitude of values, while spatial pattern correlation captures the fidelity of the overall spatial distribution, which is particularly important in sub-seasonal forecasting. All configurations were selected through trial-and-error experiments to ensure training stability and generalization capability. These details have been incorporated into the main manuscript for transparency.

The authors also select 16 sub-variables from the ECMWF forecasts and split them into two groups, one for each target variable. But it's not clear why these specific 16 were chosen in the first place. Also, Table S1 is important for understanding this part of the method and would be better placed in the main manuscript instead of the supporting information.

Thank you for your valuable comment. Although the ECMWF forecast model provides a large number of variables, it is not practical to test them all. Therefore, we first selected two target variables: temperature and precipitation. Next, assuming that the deep learning model can learn large-scale circulation patterns, we chose variables that are commonly used to represent large-scale circulation, including the zonal and meridional wind components (u, v) and geopotential height (z) at four vertical levels: near-surface (10 m u, v and mean sea level pressure), lower troposphere (850 hPa), mid-troposphere (500 hPa), and upper troposphere (200 hPa). Total column water (TCW) was included to represent atmospheric rivers, which are important for precipitation over the western United States, and elevation was used because it is known to help improve temperature bias correction (Rasp and Lerch, 2018). We have added a discussion on these selections in the manuscript and, as per your suggestion, moved Table S1 to the main text to improve clarity.
Line 122: This includes variables representing large-scale circulation at four vertical levels: near surface, lower, mid, and upper troposphere. The tcw was included to capture atmospheric rivers affecting precipitation in the western US. Elevation was included for its known benefit in temperature bias correction (Rasp and Lerch, 2018).

I commend the authors for choosing this train/validation/test split well. The three datasets are kept independent, which is good and as it should be. That said, there isn't enough clear explanation of how each dataset was used. Different model configurations are being compared directly on the test set, which isn't ideal, since model selection and tuning shouldn't be done on the test set. The test set should be reserved for evaluating the final, fully trained, "best" model on truly unseen data. As it stands, no single "final" model is chosen in the paper. If this particular architecture is intended for operational use, defining one final model configuration (e.g., V1 with E50M) with fixed inputs would make sense. This final model should also be explicitly mentioned in the conclusion, or even the abstract, along with its performance metrics compared to the baseline. That would help demonstrate more clearly that the U-Net model offers meaningful improvements and is worth using.

Thank you for your valuable suggestion. In addition to developing the model architecture, our goal was to propose the optimal input dataset for deep learning-based post-processing. To clarify, we have revised the manuscript to provide a more detailed description of the twelve datasets used. Specifically, we trained twelve separate models using a single architecture but different input datasets. Among these, the most efficient model was found to be the V1 with E50M. We have now explicitly identified this model as the final model in both the abstract and conclusion, highlighting its improved performance compared to the NWP. This addition aims to demonstrate clearly that the U-Net model offers meaningful improvements and practical utility.
Line 10: The model using the ensemble mean and only the target variables was most efficient. This model improved the pattern correlation coefficient for temperature and precipitation by 0.12 and 0.19, respectively, over a 32-day lead time.
Line 287: Overall, the most efficient model was the ensemble average using only the target variables (E50M V1), and improvements were confirmed across all evaluation metrics except for the $E_{pre}$ index for precipitation. In particular, at a 32-day lead time, temperature and precipitation showed increases of 0.12 and 0.18, respectively, in the pattern correlation coefficient compared to NWP, along with reductions of approximately 31% and 22% in RMSE.

On a related note, it would be interesting to evaluate model performance across different climate zones in a more detailed way than just at the county level. Grouping results by LCZs or land cover types could add useful insights. I'm also missing some basic plots that would help the reader understand the study area better, such as an orography map, a land cover map, or a map showing the selected counties. Including these would improve clarity in the manuscript.

Thank you for your constructive suggestion. We conducted additional analysis using the National Land Cover Database (NLCD) to evaluate model performance according to land cover types at a 0.25° resolution for the target region. In the western United States, the area is predominantly covered by three land cover types: Shrub/Scrub(47.8%), Evergreen Forest (21.17%), and Grassland/Herbaceous(12.70%), which together account for over 80% of the region. Therefore, we focused our analysis on these three types. The results showed consistent patterns of model performance across all three land cover categories. These figures have been added to the supplementary information, and the main text has been updated to include a discussion of these findings.

Line 171: Additionally, we analyzed model performance by land cover type using the National Land Cover Database (NLCD). The western United States is dominated by three land cover classes, Shrub/Scrub, Evergreen Forest, and Grassland/Herbaceous, which collectively cover over 80 percent of the study area (Fig.S6). Our analysis focused on these classes and found consistent performance patterns across all three (Figs.S7 and S8).

[Figure]

Figure R1: (Figure S6) Land cover classification of the study area using the National Land Cover Database (NLCD).

[Figure]

Figure R2: (Figure S7) Benchmark scores for precipitation forecasts, similar to Fig. R5 but focusing on three major land cover classes: Shrub/Scrub, Evergreen Forest, and Grassland/Herbaceous.

[Figure]

Figure R3: (Figure S8) Same as Fig. R2, but for temperature forecasts.

For precipitation, it's important to understand the local climate of the area being studied. Is it generally dry, or does it experience frequent rainfall? A basic analysis of the weather types, or even some box plots showing the precipitation distribution over the training set, could help contextualize things. Additionally, removing some dry days from the dataset might improve the precipitation model by helping to balance the data, giving more weight to the days when precipitation occurs. It is unclear whether the authors considered this.

Thank you for your constructive suggestion. We have also been aware of this issue. The precipitation data are indeed highly skewed, exhibiting a distribution similar to that shown in Figure R. While the preprocessing method we applied, based on approaches from selected previous studies (Aich et al.,2024), may not fully resolve this skewness, it was considered preferable to using the raw data directly. Additionally, we performed a seasonal analysis to investigate potential biases in specific climates, such as California, where precipitation is high in winter but minimal in summer. The results show improvements in all performance indices, except for precipitation $E_{pre}$, across all seasons. For precipitation $E_{pre}$, performance decreased in summer and fall but increased in winter and spring. Based on these results, we conclude that seasonal bias is minimal. We have added these figures to the supplementary materials and included a discussion of these findings in the main text.

Line 167: Before conducting a detailed analysis of the results, we examined the potential for seasonal bias and the performance by land cover type. Our findings show improvements in all seasonal evaluation metrics for both temperature and precipitation, except for precipitation $E_{pre}$ in spring and summer (Figs.S4 and S5). This suggests that the enhanced performance is not simply due to the model converging toward the seasonal mean across all seasons. Rather,

the improvements reflect the model's ability to capture relevant patterns within each season.

[Figure]

Figure R4: PRISM precipitation distributions of the training set. (a) shows the distribution of raw values, and (b) shows the distribution after pre-processing.

There's a mismatch between the overall performance of the precipitation and temperature U-Net models when comparing the full spatio-temporal test set to the specific county-level results. In many of the plots in Fig. 7 & 8 (especially for precipitation), the U-Net models do to perform significantly better than the baseline, particularly when looking at the RMSE scores. This discrepancy deserves a more thorough explanation, if not a dedicated analysis, to better understand where the model performs well and falls short. It might help to break down the results by land cover categories or orographic features over the entire spatial domain. Additionally, to back up your claims, you could run a statistical analysis to test whether the differences between models and baselines are significant, using p-values rather than just examining the plots.

Thank you for your suggestion. At the county level, the RMSE performance is similar to that of the NWP baseline. This is in discrepancy to the full spatial regional results. This is likely because the county-level regions are much smaller compared to the 1.5° resolution of the input data, which increases uncertainty and can make improvements appear less pronounced. Furthermore, when we combined and masked the developed land cover classes (categories 21–24) and re-evaluated the results (Fig. R9), we found that a sufficiently large number of grids led to noticeable improvements, supporting the interpretation that the small region size was a key factor. We have revised the text to clarify this point. In addition, following your advice, we conducted an additional statistical analysis to examine whether the performance differences between each model and the NWP baseline are statistically significant, and we have included these results in the manuscript.
Line 162: The 3D U-Net consistently outperformed the raw NWP forecasts across three evaluation metrics, except for $E_{pre}$ in precipitation. Statistical tests comparing each model's evaluation metrics with those of the NWP baseline showed that, apart from the $E_{pre}$ metric for precipitation, the improvements were significant. For precipitation $E_{pre}$, the results were mixed: five models (E01 V4, E50 V2, E50 V8, E50M V1, and E50M V2) showed no significant improvement, while seven models exhibited significant degradation.
Line 263: Note that the performance differences among 3D U-Net configurations for both targets are generally small at this county scale, while not identical to the patterns observed at larger spatial scales. This may be partly due to the very small size of the counties, which can

increase uncertainty in the evaluation. As suggested by the land cover analysis, including a sufficiently large number of grids makes performance improvements more apparent, implying that the limited spatial coverage may have constrained the observed benefits.

[Figure]

Figure R5: Benchmark scores for the Western U.S., comparing NWP and 3D U-Net models for temperature and precipitation forecasts over 32 days on four 'developed' land cover classes in Fig. R1. Layout is the same as in Fig. 3.

As a final remark, I have some uncertainties about the scope of the paper. The choice to focus only on the West Coast of the USA isn't fully explained. Since PRISM covers the entire continental U.S., wouldn't it be interesting to train the model on different patches across the country? That way, you could capture various terrains and weather types, making the model more generalizable. If the goal is to stick to the West Coast only, then this point doesn't apply. But if this is meant as a trial before scaling up, testing the model on unseen regions (maybe even using a different validation set) would be useful to see how well it performs in entirely new terrain.

Thank you for your valuable suggestion. We selected the western United States due to its complex terrain, which provide a challenging environment for sub-seasonal forecasting. Also, given the memory constraints of the 3D U-Net architecture and our current computational resources, we limited the spatial domain to this region. Nonetheless, we recognize the importance of testing model generalization across diverse areas. With improved hardware in the future, we plan to expand our approach to include multiple regions across the continental U.S., allowing us to evaluate performance on different terrains and unseen areas. We appreciate your insightful feedback and will consider it in our future work.

To summarize, the paper presents a solid application of 3D U-Net for sub-seasonal postprocessing, with some novel and practical insights, particularly in the experiments involving ensemble configurations and input variables. That said, the overall contribution would be strengthened by more thorough justification of architectural choices, more detailed methodological explanations, and a deeper critical analysis of the model's performance limitations.

Thank you very much for your valuable comments and constructive suggestions. We have addressed all your points thoroughly and carefully in the revised manuscript and our responses.

**Line-by-line and additional smaller comments:**

**0) Abstract:**

- Consider mentioning a value for one of the error metrics in the abstract, such as RMSE, to give an idea of how good or bad your models are.

Thank you for your feedback. We added the pattern correlation coefficient in the abstract.
Line 10: The model using the ensemble mean and only the target variables was most efficient. This model improved the pattern correlation coefficient for temperature and precipitation by 0.12 and 0.19, respectively, over a 32-day lead time.

- line 5: Using the ECMWF ensemble forecasting system (input) and high-resolution PRISM data (target), we tested different combinations of ensemble members and meteorological variables ...

Thank you. We have revised the sentence accordingly.

**1) Introduction**

- line 23: The U-Net architecture has been widely utilized for weather ...

Thanks for the suggestion. We've updated the sentence.

- line 32: based on a U-net model ...

Thank you. The sentence has been revised.

- line 33: The author could clarify what "only use target variables" means. Give a one-sentence explanation that you use the same variable as input (e.g., ECMWF precipitation) as your target (PRISM precipitation).

We appreciate your comment. The sentence has been added.
Line 35: Some studies use only target variables, meaning the same variable is used as both input and target, such as using ECMWF precipitation as input and PRISM precipitation as the target (Xin et al., 2024),

- line 43: Our 3D U-net architecture...

- line 43: Explain what the 3D part of the 3D U-net model means.

Thanks. The sentence has been modified as suggested.
Line 50: our 3D U-Net architecture which uses three-dimensional convolution to capture spatial and temporal features,

**2) data and method**

Line 44: Data and Methodology or Data and Methods instead of Data and Method.

Thank you. The section title has been revised.

Line 49: Give a rough approximation of the grid resolution in km x km for your area of interest.

Thanks for the suggestion. We've updated the sentence.
Line 57: we select the $1.5° \times 1.5°$ resolution (approximately 120 km $\times$ 120 km over the study region),

Line 50: Mention the temporal resolution of the ECMWF dataset.

Thank you for the feedback.
Line 59: The 2m temperature and total column water are provided as daily averaged, while the other variables are available with 6-hourly frequency.

Line 51: Also mention which variables are present within this data set, optionally put that in supporting info.

We appreciate your comment. We've updated the support info.

(Table S1) Summary of the datasets used in this study.

| Dataset | Type | Period | Resolution | Variable | Frequency |
|---------|------|--------|------------|----------|-----------|
| ECMWF | forecast | 2015-2023 | 1.5° | t2m, tcw
u, v, z, mslp, pr | daily
6-hourly |
| PRISM | reanalysis | 2015-2023 | 0.042° | t2m, tcw, mslp, pr, topo, u, v, z | daily |

Line 55, what variables are present within the PRISM data set? Put it in the support info as well.

Thanks for the suggestion. We've updated the sentence and support info.
Line 66: PRISM offers grid estimates of including temperature, precipitation, and elevation, variables at a fine spatial resolution of $0.042° \times 0.042°$ (approximately 4 km).

Line 61: Add a figure where you mark these counties on the map. It is unclear where these areas are situated on the map.

Thank you for the feedback. We add the figure in the support info (Fig.S1).

Line 67: The U-net architecture ...

Thanks, we corrected it.

Line 79: Avoid using ";"

Thank you for feedback. We corrected the semicolon to a colon as suggested.

Line 80: "merely exploiting the mean" is an awkward phrasing, rephrase it.

We appreciate your comment. The sentence has been revised.
Line 96: Using only the first ensemble member (E01), utilizing all 50 ensemble members (E50), and employing the mean of all 50 ensemble members (E50M).

Line 89: The adam-optimizer...

Thanks. The sentence has been corrected.

Line 89: Add the missing information on how you trained your model in the support information (batch size, epochs, methods to combat overfitting ...)

Thank you for pointing this out. We have updated the manuscript to include this information, and it is addressed in the response to the general comment.

Figure 1: This figure is a bit too simplistic. Consider making a more detailed figure for the model architecture. See the Ronneberger paper for reference.

Thank you for your feedback. We updated our schematic diagram.

Line 91: Use abbreviations for all of the variables for consistency.

Thank you for the detailed feedback. We added abbreviations for all variables.

Line 97: Why a 0.25° x 0.25° grid. Do you do the same for PRISM data or only the ECMWF data?

Thank you for your feedback. Due to computational limitations, we used a 0.25° grid as a compromise. PRISM data were downscaled from 0.042° to 0.25°, and ECMWF forecasts were upscaled from 1.5° to 0.25° before being input into the model. This clarification has been added to the manuscript.
Line 130: All datasets were interpolated to the $0.25° \times 0.25°$ latitude-longitude grid for model input, with PRISM data downscaled from $0.042° \times 0.042°$ and ECMWF forecasts upscaled from $1.5° \times 1.5°$.

Line 100: Would recursive feature elimination not be a more robust way to select your features, or a different feature-importance method, rather than just correlation? Does this not potentially eliminate features that have a non-linear interaction with the target variables?

Thank you for your insightful comment. We agree that recursive feature elimination or other feature-importance–based methods could provide a more robust feature selection process, particularly for capturing potential non-linear interactions with the target variables. In this study, to select the appropriate dataset, we used a simple method of choosing variables based on the correlation between the mean field and the target variable, following the relationship between the mean state and predictability described in sub-seasonal prediction studies (Lee et al., 2010; Ryu et al., 2024). We acknowledge that this approach may overlook variables with weak linear correlations but strong non-linear relationships with the target. We will consider the method you suggested in future studies to address this limitation.

Line 102: Add one extra line clarifying why this transformation (Aich et al., 2024) is needed.

Thank you for your comment. We have added a sentence to clarify the reason for applying the transformation described in Aich et al. (2024). This transformation, which includes adding a small constant, applying a log base-10 transformation, and standardizing, is designed to compress the wide range of precipitation values and stabilize variance. Following this approach, Aich et al. (2024) successfully performed precipitation bias correction and high-resolution downscaling, which motivated us to adopt the same method in our study. This explanation has been added to the main text.
Line 137: For precipitation and tcw, Following Aich et al. (2024), we applied a transformation to compress the wide range of precipitation values and facilitate stable, efficient model training.

Line 107: Why not train on residue (Prism – ECMWF) rather than just temperature as your target?

We thank the reviewer for this helpful suggestion. In the present study, we trained the model directly on the target variable values. We agree that using the residuals (PRISM – ECMWF) as the target could be an effective alternative for bias correction. While this approach was not implemented in the current work, we will certainly consider it in future studies.

Line 110: Mention the formulas for RMSE and pattern correlation in the supporting information.

Thank you for your feedback. We have added the formulas for RMSE and pattern correlation, including latitude weighting, to the supporting information.
Supporting information:

**Definition of Pattern Correlation and Root Mean Square Error (RMSE)**

$$\text{Pattern Correlation} := \frac{\sum_{i=1}^{n} w_i \left(f_i - \overline{f}\right)(o_i - \overline{o})}{\sqrt{\sum_{i=1}^{n} w_i \left(f_i - \overline{f}\right)^2} \sqrt{\sum_{i=1}^{n} w_i \left(o_i - \overline{o}\right)^2}} \tag{1}$$

$$\text{RMSE} := \sqrt{\frac{\sum_{i=1}^{n} w_i \left(f_i - o_i\right)^2}{\sum_{i=1}^{n} w_i}} \tag{2}$$

where

$$f_i : \text{ the model forecast at grid cell } i$$

$$o_i : \text{ the reference value (reanalysis) at grid cell } i$$

$$w_i = \cos(\phi_i) : \text{ latitude-based weight for the grid cell at latitude } \phi_i$$

$$\overline{f} = \frac{\sum_{i=1}^{n} w_i f_i}{\sum_{i=1}^{n} w_i} : \text{ latitude-weighted means of the forecast}$$

$$\overline{o} = \frac{\sum_{i=1}^{n} w_i o_i}{\sum_{i=1}^{n} w_i} : \text{ latitude-weighted means of reference fields}$$

Line 110: Avoid using ";"

Thank you for your feedback. Corrected the semicolon to a colon as suggested.

**3) Results & discussion**

Figure 2: Why is the RMSE & pattern correlation of the U-net model so much better, but the E_pre is more on par with the baseline for precipitation, and why is it not the case for temperature? If you were to choose your best model, based on which metric would you do that?

We appreciate your comment. This is because the U-Net model shows superior performance in spatial pattern correlation and RMSE. However, for precipitation, it generally underestimates values, as shown in Figure 4. This underestimation leads to reduced variance, which, when combined with improved spatial patterns, results in $E_{pre}$ values that are comparable to the baseline. We have consistently noted this underestimation issue for precipitation throughout the manuscript. In selecting the best model, we considered all three metrics equally and chose the model that achieved the largest overall improvement across them. As a result, E50M V1 was selected, and this has been explicitly stated in the main text.
Line 287: Overall, the most efficient model was the ensemble average using only the target variables (E50M V1), and improvements were confirmed across all evaluation metrics except for the $E_{pre}$ index for precipitation. In particular, at a 32-day lead time, temperature and precipitation showed increases of 0.12 and 0.18, respectively, in the pattern correlation coefficient compared to NWP, along with reductions of approximately 31% and 22% in RMSE.

Line 132: Avoid using ";"

Thank you for feedback. Corrected the semicolon to a colon as suggested.

Line 135: Also, what sub-variables were used to obtain these figures? Mention these here or in the support information.

Thank you for your comment. We conducted a total of 12 experimental configurations, and Figures 2 and 3 present the average results of experiments under the same conditions. For example, in Figure 2, the "E50" results represent the average of E50 V1, E50 V2, E50 V4, and E50 V8, while in Figure 3, the "V2" results represent the average of

E01 V2, E50 V2, and E50M V2. We have revised the main text to clearly state this information.
Line 175: We then grouped experiments with the same input variables and ensemble configurations to assess the role of auxiliary variables and ensemble structure, for example, averaging E50 V1, E50 V2, E50 V4, and E50 V8 for the E50 group.

Line 138: Put the mentioned figure in the support information.

Thank you for your feedback. We have added the results of the additional experiment to the supplementary information. While the main experiments were trained with PRISM data and tested on 2023 forecasts, the supplementary figure presents results from a complementary experiment trained with ERA5 data and tested on 2022 forecasts. This additional experiment supports the finding that the performance difference between E50 and E50M is negligible, as ensembles of 10 and 20 members also yielded similar results to E50.
Line 184: To support these findings, we conducted a complementary experiment trained and teested with ERA5 data and tested on 2022 forecasts. The results indicate that ensembles of 10 and 20 members achieved performance comparable to E50(Fig.S9).

Line 146: Your model surpasses the baseline slightly, but is it significant? Here, it would be interesting to show error bars or some statistics (p-values) to truly show that it is worth keeping V8, because including more input variables means both a longer training time and higher complexity of the model. Also, make sure to mention what you did with the ensemble members here. Were they averaged out in E50M?

Thank you for the feedback. The significance tests produced mixed results. V8 showed significant improvement compared to V2 and V4, but not compared to V1. We have revised the main text to reflect this.
Line 198: The impact of input variables on model performance is further explored in Fig. 2, which represents the averages of E01, E50, and E50M.
Line 200:Specifically, V8 shows significant improvements over V2 and V4 in all temperature metrics, but performs similarly to V1.

Figure 4 & Figure 5: There are a lot of plots, and not all say equally as much. Is it possible to make them smaller or use fewer graphs? For example, with the temperature plots, the first four rows are too similar to each other to show differences. Perhaps those can be put in supporting info, and you only keep the difference plots. For additional clarity, you could convert the temperatures here to degrees Celsius rather than Kelvin. Potentially add an error metric to the difference plots, such as an average error or MAE. This will make it more understandable how far the predictions are from the PRISM values.

Thank you for your comment. Following your suggestions, we updated the figures by moving raw plots to the supplementary information, retaining only the difference plots in the main text. Temperatures are now displayed in degrees Celsius, and an additional error metric (MAE) has been added to the difference plots for improved clarity.

Line 169: I suggest to adding a landcover map or something to help illustrate this better.

Thank you for your constructive suggestion. A land cover map has been added to the supplementary information, and the short discussion on land cover has also been added to main text.
Line 225: Moreover, improvements were observed across the three dominant land cover types, which together account for over 80% of the study area (Figs.S7 and S8).

Figure 7: Most models seem to be on par with the baseline for three counties (except Salt Lake City and Seattle). This seems a bit contradictory to the figures above. Also, plot the black line on top to compare your models with it, or make it more visible. Is it necessary to have all 12 models in these figures? The graph style itself is very nice, but it is a bit crowded. What about the E_pre metric for these locations?

Thank you for your suggestion. Following your advice, we have updated Figure 7 to make the baseline more visible and the comparisons clearer. As also discussed in our response to the mismatch between the overall performance and the specific county-level results comment, the results at the county level tend to show smaller improvements because the finer spatial scale increases uncertainty, making it more challenging for the model. Regarding the $E_{pre}$ metric, it was excluded because calculating $E_{pre}$ requires pattern correlation, which is not possible here due to the use of area-averaged values. This clarification has been added to the main text.
Line 246: Fig. 7 presents comprehensive performance metrics for temperature and precipitation forecasts across these 5 regions (Fig.S1), comparing NWP with the most efficient model (E50M V1) over a 32-day lead time. Results for all models are shown in Fig.S14. The $E_{pre}$ metric was excluded for county-level results because its calculation requires spatial pattern correlation, which cannot be obtained from area-averaged values.

Line 195: potential indication to use LCZ, land cover, or some urban-rural-sea mask as an input parameter.

Thank you for your suggestion. We have added a discussion on using land cover as additional input parameters in future work.
Line 255: Incorporating land cover, which is already a key input in NWP models (López-Espinoza et al., 2020), could offer additional improvements in such regions

Figure 8: Perhaps mention how many extreme weather events you had in your dataset. Some box plots of the year temperatures and precipitation may be interesting in the supporting information.

Thank you for your suggestion. Extreme events in this study were defined as the top 10% and bottom 10% of daily temperature, and the top 10% of daily precipitation, within the period from January 2023 to January 2024. This corresponds to 39 samples for each extreme category. We have also examined the distribution of temperature and precipitation for this period and added the corresponding plots to the supporting information.
Line 269: Extreme events are defined as the 39 cases corresponding to the top 10% and bottom 10% of daily temperature, and the top 10% of daily precipitation, within the period from January 2023 to January 2024. The temperature and precipitation distributions for this period are shown in Fig.S15.

[Figure]

Figure R6: (Figure S15) Distribution of (a) temperature and (b) precipitation on five regions in Fig. S1.

**4) Conclusion + Support information**

Here, I miss a mention of what combination of inputs gave you the best models for temperature and precipitation. Also, provide some scores to illustrate your point (potentially mention how much it improves on the baseline). Include an example or a clearer description of your input for the model. Add the additionally mentioned plots and information on the training processes of your U-net models.

Thank you for your feedback. We have added a description of the best model in the conclusion. Additionally, clarifications regarding model and data usage were addressed in response to previous comments and have been incorporated into the main text and supplementary materials.

Line 287: Overall, the most efficient model was the ensemble average using only the target variables (E50M V1), and improvements were confirmed across all evaluation metrics except for the $E_{pre}$ index for precipitation. In particular, at a 32-day lead time, temperature and precipitation showed increases of 0.12 and 0.18, respectively, in the pattern correlation coefficient compared to NWP, along with reductions of approximately 31% and 22% in RMSE.

Citation: https://doi.org/10.5194/egusphere-2025-308-RC2